# Oxytocin neurons mediate the effect of social isolation via the VTA circuits

Stefano Musardo[1], Alessandro Contestabile[1], Marit Knoop[2], Olivier Baud[2,3], Camilla Bellone[1]*

[1]Department of Basic Neuroscience, University of Geneva, Geneva, Switzerland; [2]Laboratory of Child Growth and Development, University of Geneva, Geneva, Switzerland; [3]Division of Neonatology and Pediatric Intensive Care, Children's University Hospital of Geneva, Geneva, Switzerland

**Abstract** Social interaction during adolescence strongly influences brain function and behavior, and the recent pandemic has emphasized the devastating effect of social distancing on mental health. While accumulating evidence has shown the importance of the reward system in encoding specific aspects of social interaction, the consequences of social isolation on the reward system and the development of social skills later in adulthood are still largely unknown. Here, we found that 1 week of social isolation during adolescence in male mice increased social interaction at the expense of social habituation and social novelty preference. Behavioral changes were accompanied by the acute hyperexcitability of putative dopamine (pDA) neurons in the ventral tegmental area and long-lasting expression of GluA2-lacking AMPARs at excitatory inputs onto pDA neurons that project to the prefrontal cortex. Social isolation-dependent behavioral deficits and changes in neural activity and synaptic plasticity were reversed by chemogenetic inhibition of oxytocin neurons in the paraventricular nucleus of the hypothalamus. These results demonstrate that social isolation in male mice has acute and long-lasting effects on social interaction and suggest that homeostatic adaptations mediate these effects within the reward circuit.

*For correspondence:
Camilla.Bellone@unige.ch

Competing interest: The authors declare that no competing interests exist.

## Editor's evaluation

This paper evaluated the lasting effects of acute social isolation on future social interactions in juvenile mice, revealing a compelling oxytocin-mediated mechanism. A clear hypothesis has been laid out within a defined anatomical framework, and social interactions were evaluated using appropriate behavioral paradigms, chemogenetic, and pharmacological tools. The work provides new insights on oxytocin signaling as a key regulator of the neural substrates underlying enduring effects of social interaction.

## Introduction

The experience of social interaction during postnatal development and adolescence is fundamental for setting the basis for social life, and the deprivation of social experience (hereby defined social isolation) impacts the survival of all species, as suggested by the adverse effects of the massive social isolation imposed by the COVID-19 crisis on mental health (*Pancani et al., 2021*). Identifying the possible neural mechanisms underlying the negative consequences of social isolation may help prevent and treat mental disorders.

In rodents, depending on the duration of juvenile social isolation, increased or decreased sociability in adulthood has been reported, suggesting that adolescence is a sensitive period for establishing social behavior later in life (*Yamamuro et al., 2020*; *Yamamuro et al., 2018*; *Makinodan et al.,*

*2012*; *Rivera-Irizarry et al., 2020*). Few studies have examined the acute effects of social isolation and the cellular and circuit mechanisms that regulate the long-lasting impacts of social isolation. For example, short-term isolation has increased social interaction in rats (*Niesink and van Ree, 1982*).

Social interaction is generally a rewarding experience with reinforcing properties. Recent studies have highlighted the necessity and sufficiency of dopamine (DA) neurons of the ventral tegmental area (VTA) to promote social interaction (*Gunaydin et al., 2014*; *Solié et al., 2022*). Brief periods of acute social isolation have been reported to activate midbrain regions in humans (*Tomova et al., 2020*) and increase DA neurons' activity within the dorsal Raphe nucleus (DRN) in mice (*Matthews et al., 2016*). While in rodents, 24 hr of social isolation does not change synaptic strength onto DA neurons of the VTA (*Matthews et al., 2016*), in humans, the response of the VTA to social cues after brief isolation is increased (*Tomova et al., 2020*). These data suggest that changes within VTA DA neurons may be the substrate for social craving caused by acute social isolation. The neuronal mechanisms and long-term consequences remain uninvestigated.

DA neurons of the VTA contribute to reward-seeking behavior, motivation, and reinforcement learning. Their activity is controlled upstream by several brain structures (*Tomova et al., 2020*), each of which may contribute to distinct behavioral aspects. DA neuron activity is controlled by glutamatergic and GABAergic inputs and tightly regulated by neuromodulators that act on G protein-coupled receptors (GPCRs). Oxytocin is a neuropeptide released by neurons within the paraventricular nucleus (PVN) of the hypothalamus that directly projects to the VTA. Oxytocin in the VTA activates oxytocin receptors on DA neurons regulates their activity, suggesting that oxytocin in the VTA gates social reward (*Hung et al., 2017*; *Tang et al., 2014*; *Xiao et al., 2017*). Indeed, the presence of a conspecific activates the oxytocin system, which increases oxytocin release, activates DA neurons of the VTA, and promotes the initiation and maintenance of social interaction (*Resendez et al., 2020*; *Oettl et al., 2016*). Although it has been hypothesized that oxytocin senses changes in the environment and facilitates behavioral stability to better adapt to changes (*Quintana and Guastella, 2020*), the role of oxytocin neurons in the behavioral consequences of social isolation remains largely unknown.

## Results

We first characterized the acute consequences of short-term social isolation on social interaction during adolescence. Male mice were weaned at postnatal day (P) 21 and then isolated between P28 and P35 (*Figure 1A*). On the last day, we exposed the experimental mice to an unknown sex-matched juvenile conspecific or a novel object in a direct free-interaction task (*Figure 1B*). Isolated mice spent more time interacting with the conspecific than the grouped control mice (*Figure 1C*). Conversely, object exploration did not differ between the two groups, indicating that social isolation preferentially affects social exploration (*Figure 1B, D*). Increased social interaction was also found when a former cage mate was presented as a social stimulus (*Figure 1E, F*). At the same time, it was not observed after a brief 24-hr period of social isolation, as expected from previous work (*Matthews et al., 2016*; *Figure 1—figure supplement 1A, B*). These data suggest that the duration of the isolation is an essential factor in determining the behavioral consequences of social isolation.

To further investigate the consequences of social isolation on different aspects of social behavior, we used a three-chamber interaction task to characterize sociability and social novelty preference (*Figure 1G*). We found that the socially isolated and grouped mice spent significantly more time investigating a juvenile conspecific over a novel object (*Figure 1H and* – supplement 1 D, E, F). In the second part of the test, in contrast to the control mice, the socially isolated mice spent the same amount of time interacting with familiar and unknown conspecifics (*Figure 1I and* – supplement 1 G, H), indicating that social isolation affects social novelty preference. After social isolation, mice presented deficits in habituation when exposed to the same juvenile conspecific for four consecutive days (*Figure 1J–N*). but not when exposed to the same juvenile conspecific for four consecutive trials on the same day (*Figure 1—figure supplement 1I-M*). Novel object recognition (*Figure 1—figure supplement 2A-F*) and behavior in the EPM (*Figure 1—figure supplement 2A, H-L*) did not differ between the socially isolated and control mice. Moreover, social isolation does not impact the reinforcing properties of the social interaction since both groups of mice preferred the social chamber when they were subjected to social conditioned place preference (sCPP—*Figure 1—figure supplement 2M-R*). To investigate whether social isolation-dependent behavioral deficits are age-dependent, we studied isolated mice during adulthood (7 days of isolation; P53–P60, *Figure 1—figure supplement 3A*). We found that

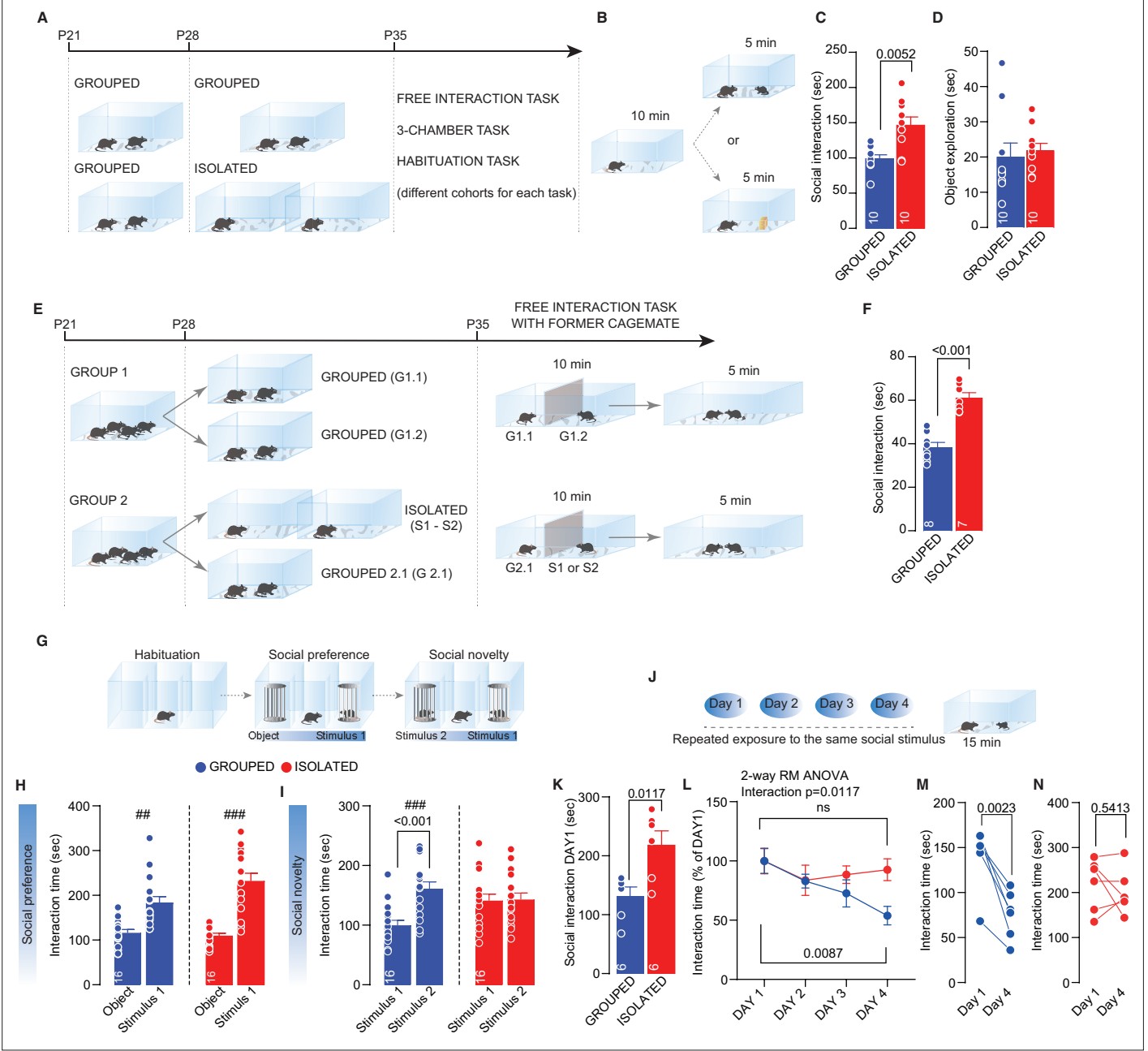

**Figure 1.** Adolescence acute social isolation induces social craving. (**A**) Experimental design: WT mice were isolated between P28 and P35 or kept in group. After isolation, mice (from different cohorts) were subjected to different behavioral task. (**B**) Free direct interaction task paradigm. (**C**) Time exploring social stimulus (Mann-Whitney U-test=14, p=0.0052, n=10 mice each group). (**D**) Time exploring object (Mann-Whitney U-test=33, p=0.2176, n=10 mice each group). (**E**) Experimental design: mice were house four per cage from P21 to P28. At P28, one group of mice were divided in two cages with two mice each, while a second group was divided in one cage with two mice and two cages with one mouse until P35. After isolation, a free-interaction with former cage mate was carried out. (**F**) Interaction time with former cage mate (Unpaired sample t-test, $t_{(13)}$=6.885, p<0.001). (**G**) Three-chamber task experimental paradigm. (**H**) Interaction time with object or social stimulus 1 (Grouped ## Wilcoxon matched-pairs signed rank test, W=118, p=0.001; Isolated ### Paired sample t-test, $t_{(15)}$=5.975, p<0.001. Two-way RM-ANOVA. Target main effect $F_{(1, 30)}$=51.47, p<0.001; House condition main effect $F_{(1, 30)}$=3.935, p=0.0565; Target × House condition $F_{(2, 60)}$=9.487, p>0.001, n=16). (**I**) Interaction time with stimulus 1 (familiar) and stimulus 2 (unfamiliar) (Grouped ### Paired sample t-test, $t_{(15)}$=5.774, p<0.001; Isolated ### Paired sample t-test, $t_{(15)}$=0.102, p=0.9201. Two-way RM-ANOVA followed by Bonferroni's multiple comparisons test. Target main effect $F_{(1, 30)}$=9.251, p=0.0049; House condition main effect $F_{(1, 30)}$=1.164, p=0.2892; Target × House condition $F_{(1, 30)}$=8.215, p=0.0075, n=16). (**J**) Habituation task paradigm. (**K**) Interaction time on Day 1 (Unpaired-samples t-test, $t_{(10)}$=3.076, p=0.0117, n=6 mice each group). (**L**) Interaction time across 4 days (Two-way RM ANOVA followed by Tukey's multiple comparisons test, Days main effect $F_{(2.215, 22.15)}$=6.775, p=0.0041, House condition main effect $F_{(1, 10)}$=1.455, p=0.2555, Days × House condition $F_{(3, 30)}$=4.349, p=0.0117, Grouped Day 1

*Figure 1 continued on next page*

*Figure 1 continued*

versus Day 4, p=0.0087, Isolated Day 1 versus Day 4, p=0.9094, n=6 mice each group). (**M**) Interaction time during Day 1 and Day 4 for Grouped mice (Paired samples t-test, $t_{(5)}$=5.706, p=0.0023). (**N**) Interaction time during Day 1 and Day 4 for isolated mice (Paired samples t-test, $t_{(5)}$=0.6552, p=0.5413). Data are represented as mean ± SEM.

The online version of this article includes the following source data and figure supplement(s) for figure 1:

**Source data 1.** *Figure 1* - raw data and statistical output.

**Figure supplement 1.** Effects of social isolation on social behavior.

**Figure supplement 1—source data 1.** *Figure 1—figure supplement 1* - raw data and statistical output.

**Figure supplement 2.** Effects of social isolation on social behavior.

**Figure supplement 2—source data 1.** *Figure 1—figure supplement 2* - raw data and statistical output.

**Figure supplement 3.** Effects of social isolation during adulthood.

**Figure supplement 3—source data 1.** *Figure 1—figure supplement 3* - raw data and statistical output.

isolated adult mice spent less time interacting with a conspecific than the control mice. At the same time, object exploration (*Figure 1—figure supplement 3B-D*), sociability, and social novelty preference (*Figure 1—figure supplement 3E-L*) were no different. Altogether, these data indicate that adolescence is a critical period for developing social behavioral skills and that social isolation during this period increases social interaction at the expense of an impaired social novelty preference and altered habituation to interact with a familiar conspecific.

In humans, acute social isolation has been reported to have a rebound effect on conspecific interaction, accompanied by an increase in the response of reward circuits and, in particular, the VTA in response to social cues (*Tomova et al., 2020*). In line with these discoveries and further investigating the related neural mechanisms, we measured the excitability of pDA in the VTA after social isolation. Mice were isolated or maintained in group housing from P28 to P35, and acute brain slices were subsequently prepared (*Figure 2A*). We observed increased excitability after social isolation without a change in the resting membrane potential (*Figure 2B and D*). Aiming to identify the upstream key

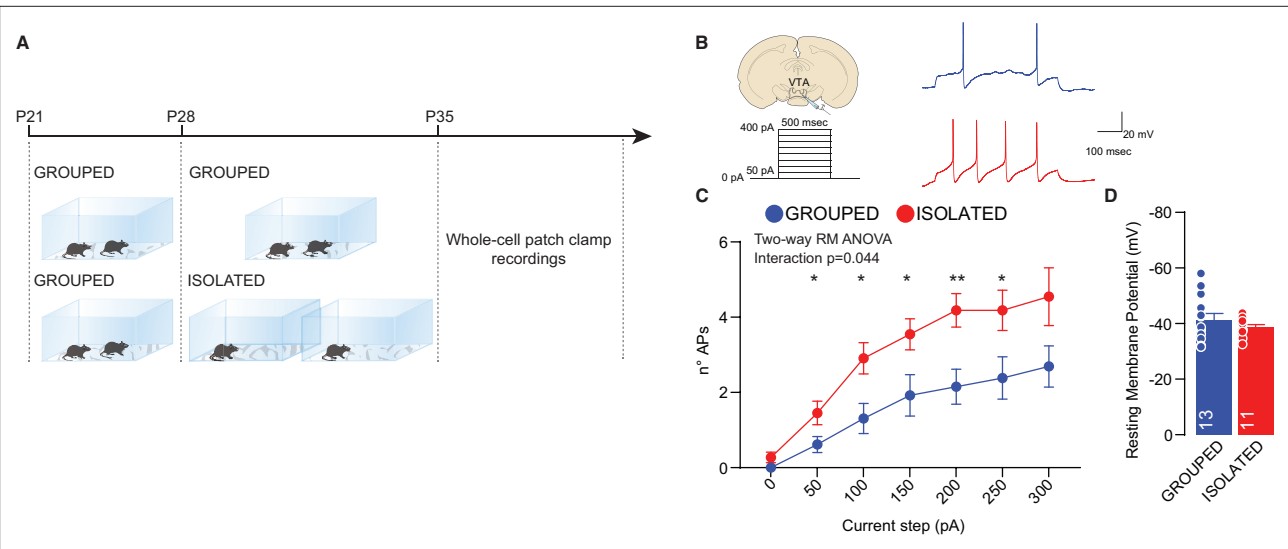

**Figure 2.** Adolescence acute social isolation induces putative VTA DA neurons hyperexcitability. (**A**) Experimental design: WT mice were isolated between P28 and P35 or kept in group. After isolation, mice were subjected whole-cell patch-clamp recordings. (**B**) Left: experimental paradigm, putative VTA DA neurons were subjected at multiple depolarizing current steps. Right: example traces from 250 pA depolarizing current injection. (**C**) Number of action potentials (APs) across increasing depolarizing current steps (Two-way RM ANOVA followed by uncorrected Fisher's LSD post doc analysis, House condition × Current step $F_{(6, 132)}$=2.230, p=0.044, current step main effect $F_{(2.310, 50.82)}$=36.65, p<0.001, House condition main effect $F_{(1, 22)}$=8.016, p=0.0097). (**D**) Resting membrane potential of recorded cells (Mann-Whitney U-test=66, p=0.7762. Grouped n=13, Isolated n=11 from three mice each group). Data are represented as mean ± SEM.

The online version of this article includes the following source data for figure 2:

**Source data 1.** *Figure 2* - raw data and statistical output.

brain regions responsible for regulating the activity of DA neurons, we performed cFos analysis of brain slices seven days after social isolation. We observed an increase in PVN neurons immunopositive for cFos, suggesting increased activity (*Figure 3A, B*, *Figure 3—figure supplement 1B*). Because of their role in social behavior and the regulation of DA neuron activity (*Hung et al., 2017*; *Xiao et al., 2017*), we then focused our analysis on oxytocin neurons within the PVN. By confocal imaging quantification, we observed an increase in the density of oxytocin neurons in the PVN (*Figure 3A, C*, *Figure 3—figure supplement 1D*) and an increase of OXT+/cFOS+ double-positive cells (*Figure 3A and D*) after social isolation. On the contrary, social isolation does not affect the number of vaso-pressin (AVP) positive cells nor the AVP+/cFOS+ or OXT+/AVP+ (*Figure 3—figure supplement 1C, E, H*), suggesting that social isolation induces an alteration of oxytocin rather than a modification of the cell identity.

PVN neurons have been shown to regulate the activity of DA neurons in the VTA; therefore, we tested the hypothesis that these neurons are the master regulator of DA neuron activity during social isolation. To test our hypothesis, mice were first injected with CTB-488 in the VTA at P21 and then isolated between P28 and P35. After isolation, we performed patch-clamp recording from PVN neurons projecting to the VTA (PVN-VTA), and we observed increased excitability compared to that of the control group (*Figure 3E–G*).

To prove the causal link between oxytocin neurons and DA neuron activity during social isolation, we crossed R26-hM4Di/mCitrine mice with Oxytocin-Ires-Cre mice to express a designer inhibitor receptor-activated exclusively by designer drugs expressed under the Cre promoter in Oxt-positive cells (*Figure 3H*). Clozapine-N-oxide was dissolved in drinking water and administered during social isolation (or in parallel to grouped control mice). We obtained whole-cell patch-clamp recordings from pDA neurons in the VTA. We observed the rescue of excitability in cells recorded from isolated mice treated with CNO but not isolated mice treated with vehicle (*Figure 3I-L*, *Figure 3—figure supplement 1I-L*). Finally, to prove causality between neuronal hyperexcitability and behavior, we treated Oxt-hM4Di mice with CNO or vehicle. We compared the time spent interacting with a novel conspecific after social isolation (*Figure 3M*). We also used grouped mice treated with CNO or vehicle as controls. As expected, we found an increase in interaction time in untreated isolated versus grouped mice, while no difference was observed between mice treated with CNO independent of housing (*Figure 3N*). Altogether, the data presented here indicate that increased social interaction after social isolation is the consequence of the increased excitability of oxytocin neurons of the PVN and suggest that this effect is mediated by the increased activity of pDA neurons within the VTA.

We next investigated whether social isolation during adolescence has long-lasting consequences and the consequent neural mechanisms. After 7 days of social isolation from P28 to P35, the mice were regrouped, and social behavior was then tested during adulthood (*Figure 4A*). We still observed increased social interaction in mice isolated during adolescence compared to the control group (*Figure 4B and C*), but object exploration was not affected (*Figure 4D*). Sociability, social novelty pref-erence (*Figure 4—figure supplement 1A-I*), and social habituation were similar between grouped and regrouped mice (*Figure 4E, I*), although during the first interaction, regrouped mice interacted more (*Figure 4F*). These behavioral data indicate that acute social isolation during adolescence leads to a long-lasting increase in free/unrestrained social interaction during adulthood. Remarkably, inhibi-tion of oxytocin neuron activity during social isolation was sufficient to block the long-lasting conse-quences of social isolation and restore social behavior (*Figure 4J–K*). These data indicate that an isolation-dependent increase in social interaction occurs during adulthood and support the possible role of oxytocin neurons in regulating social craving after isolation. To investigate the related neural mechanisms underlying the long-lasting consequence of social isolation, we performed a whole-cell patch-clamp recording of pDA neurons in the VTA.

Neuronal excitability in adulthood did not differ between isolated and control mice (*Figure 4—figure supplement 2A-I*), suggesting that the long-lasting behavioral consequences of social isolation, although induced by neuronal excitability, are exerted by a different neuronal mechanism. Neurons undergo other mechanisms of homeostatic adaptation to overall changes in neuronal activity, and many studies have reported in vivo scaling triggered by sensory manipulation and exerted by the regu-lation of calcium-permeable (CP)-AMPARs (*Goel et al., 2011*). We, therefore, characterized whether a long-lasting increase in free/unrestrained social interaction after social isolation during adolescence is accompanied by changes at the level of synaptic transmission at excitatory inputs onto pDA neurons

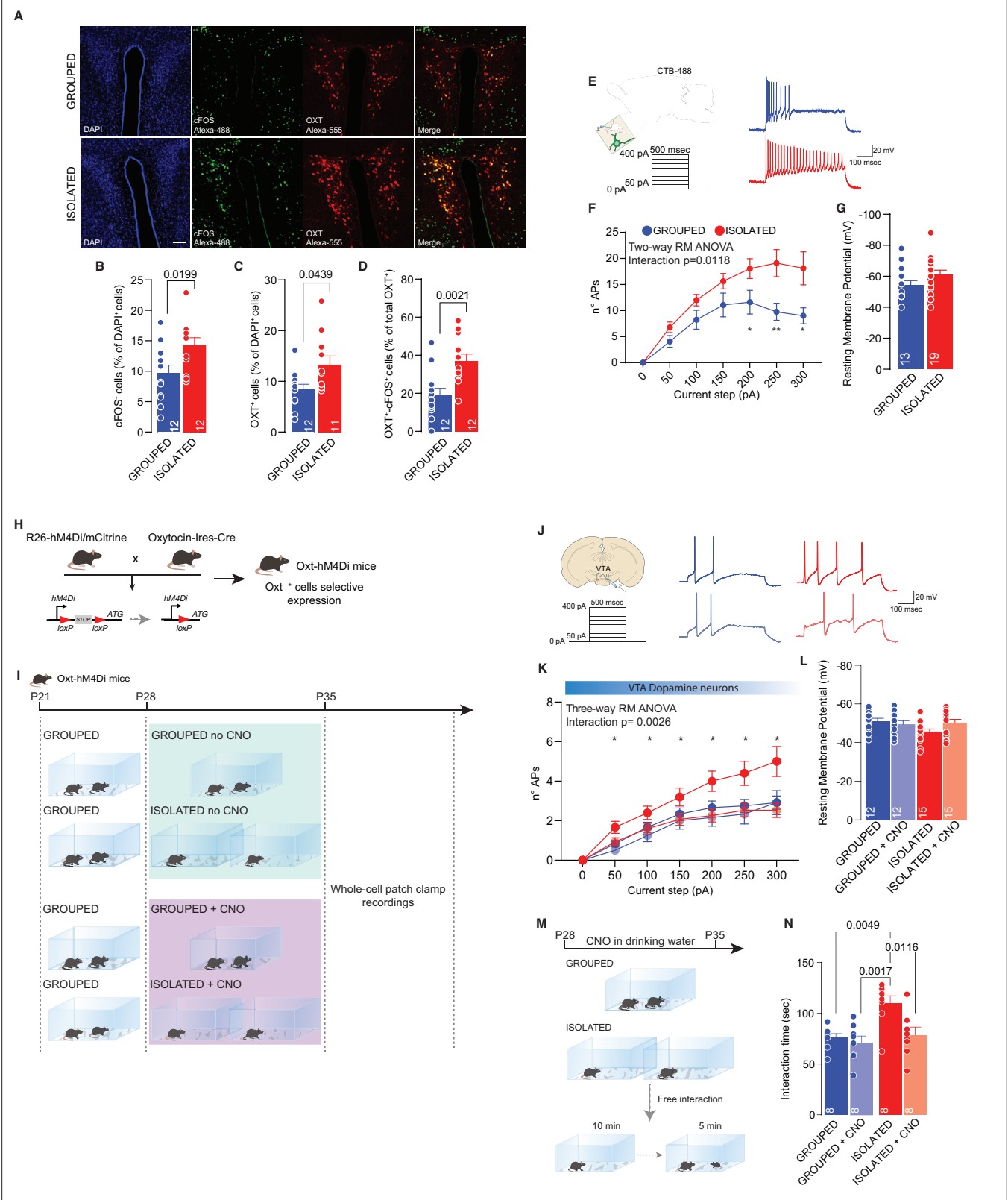

**Figure 3.** PVN OXT neurons as main orchestrator of social isolation induced social craving. (**A**) Representative confocal images of PVN stained with cFOS, OXT, and AVP antibody (scale bar: 20 µm). (**B**) cFos+ cells (as % of DAPI+ cells) (Unpaired samples t-test, $t_{(22)}$=2.510, p=0.0199, n=12 slices from three mouse each group). (**C**) OXT+ cells (as % of DAPI+ cells) (Mann-Whitney U-test=33, p=0.0439, n=11–12 slices from three mouse each group). (**D**) OXT+/ cFos+ cells (as % of total OXT+ cells) (Unpaired samples t-test, $t_{(22)}$=3.491, p=0.021, n=12 slices from three mouse each group). (**E**) Left:

*Figure 3 continued*

experimental paradigm, PVN neurons projecting to VTA (CTB-488 was injected in the VTA at P21) were subjected at multiple depolarizing current steps. Right: example traces from 250 pA depolarizing current injection. (**F**) Number of action potentials (APs) across increasing depolarizing current steps (Two-way RM ANOVA followed by uncorrected Fisher's LSD post doc analysis, House condition main effect $F_{(1, 30)}$=7.306, p=0.0112, Current step main effect $F_{(1.857, 55.71)}$=30.47, p<0.001, House condition × Current step $F_{(6, 180)}$=2.828, p=0.018). (**G**) Resting membrane potential of recorded cells (Mann-Whitney U-test=85.5, p=0.1483. Grouped n=13, Isolated n=19 from three mice each group). (**H**) Experimental design. R26-hM4Di/mCitrine mice were crossed with Oxytocin-Ires-Cre mice generating Oxt-hM4Di mice which express inhibitory DREAAD specifically in OXT neurons. (**I**) Experimental paradigm: Oxt-hM4Di mice were isolated between P28 and P35 and CNO was dissolved in the drinking water. After isolation mice were subjected whole-cell patch-clamp recordings. (**J**) Left: experimental paradigm, VTA pDA neurons were subjected at multiple depolarizing current steps. Right: example traces from 250 pA depolarizing current injection. (**K**) Number of APs across increasing depolarizing current steps (Three-way RM ANOVA followed by uncorrected Fisher's LSD post doc analysis (see *Figure 3—source data 1* for details), Current step main effect $F_{(1.962, 98.08)}$=114.8, p<0.0001, House condition main effect $F_{(1, 50)}$=3.456, p=0.0689, CNO treatment main effect $F_{(1, 50)}$=5.826, p=0.0195, Current step × House condition $F_{(6, 300)}$=1.866, p=0.0865, Current step × CNO treatment $F_{(6, 300)}$=3.806, p=0.0011, House condition × CNO treatment $F_{(1, 50)}$=2.293, p=0.1363, Current step × House condition × CNO treatment $F_{(6, 300)}$=3.459, p=0.0026). (**L**) Resting membrane potential of recorded cells (Two-way ANOVA, CNO treatment main effect $F_{(1, 50)}$=0.8745, p=0.3542, House condition main effect $F_{(1, 50)}$=1.956, p=0.1681, CNO treatment × House condition $F_{(1, 50)}$=3.531, p=0.0661, Grouped n=12, Grouped +CNO n=12, Isolated n=15, Isolated+CNO n=15 from 3 to 4 mice each group). (**M**) Experimental design. Oxt-hM4Di mice were isolated or kept grouped from P28 to P35. CNO was dissolved in drinking water and after isolation mice underwent to free direct social interaction task. (**N**) Social interaction time (Two-way ANOVA followed by Tukey's multiple comparisons test, CNO treatment main effect $F_{(1, 28)}$=7.088, p=0.0127, House condition main effect $F_{(1, 28)}$=9.940, p=0.0038 CNO treatment × House condition $F_{(1, 28)}$=4.334, p=0.0466, n=8 each group). Data are represented as mean ± SEM.

The online version of this article includes the following source data and figure supplement(s) for figure 3:

**Source data 1.** *Figure 3* - raw data and statistical output.

**Figure supplement 1.** CNO validation.

**Figure supplement 1—source data 1.** *Figure 3—figure supplement 1* - raw data and statistical output.

in the VTA. We obtained whole-cell patch-clamp recordings from pDA neurons while pharmacologically isolating excitatory transmission. Considering the output-dependent heterogeneity of DA neurons and the previously identified neuronal type specificity in the form of experience-dependent synaptic plasticity (*Bariselli et al., 2016a*; *Saal et al., 2003*), we decided to characterize long-lasting, isolation-dependent effects on the synaptic plasticity of pDA neurons in the VTA in an output-specific manner. To that end, we injected choleratoxin in the prefrontal cortex (PFC) or nucleus accumbens (NAc). We then obtained whole-cell patch-clamp recordings from the identified neuronal population (*Figure 5A, B and G*). While we observed no difference in the ratio of AMPAR- and NMDAR-mediated currents between control and isolated mice (*Figure 5C and D*), we observed an increase in rectification index (RI) at excitatory inputs onto pDA neurons projecting to the PFC in isolated mice (*Figure 5E and F*). No change in the AMPA/NMDA ratio or the RI was observed at excitatory inputs onto pDA neurons projecting to the NAc (*Figure 5H–K*). These results led us to verify whether synaptic plasticity was already present after social isolation or if, on the contrary, it represented a specific adaptive mechanism occurring in adulthood. We injected CTB into the NAc or mPFC at P21, isolated the mice between P28 to P35, and performed whole-cell patch-clamp recordings at the end of the isolation (*Figure 5—figure supplement 1A*). While we observed no difference in the ratio of AMPAR- and NMDAR-mediated currents between control and isolated mice (*Figure 5—figure supplement 1B-D*), we observed that RI was higher in isolated compared to the control group at excitatory inputs onto pDA neurons projecting to the PFC (*Figure 5—figure supplement 1E-F*). No change in the AMPA/NMDA ratio or the RI was observed at excitatory inputs onto pDA neurons projecting to the NAc (*Figure 5—figure supplement 1G-K*). Moreover, no differences between groups were observed in the spontaneous inhibitory postsynaptic currents (sIPSCs) (*Figure 5—figure supplement 1L-P*), suggesting that synaptic scaling was not the consequence of changes in inhibitory transmission.

We then tested whether the expression of CP-AMPARs on DA neurons projecting to the PFC in adulthood is the consequence of the increased excitability of oxytocin neurons in the PVN during social isolation. We isolated Oxt-hM4Di mice during adolescence, treated them with CNO or vehicle during isolation, regrouped them after 7 days until adulthood, and recorded excitatory transmission from pDA neurons projecting to the PFC from acute VTA slices (*Figure 5L*). Notably, we observed that RI was normalized when the activity of the oxytocin neurons was chemogenetically reduced during social isolation (*Figure 5M and N*). These data indicate that social isolation during adolescence leads to long-lasting effects in free/unrestrained social interaction accompanied by oxytocin

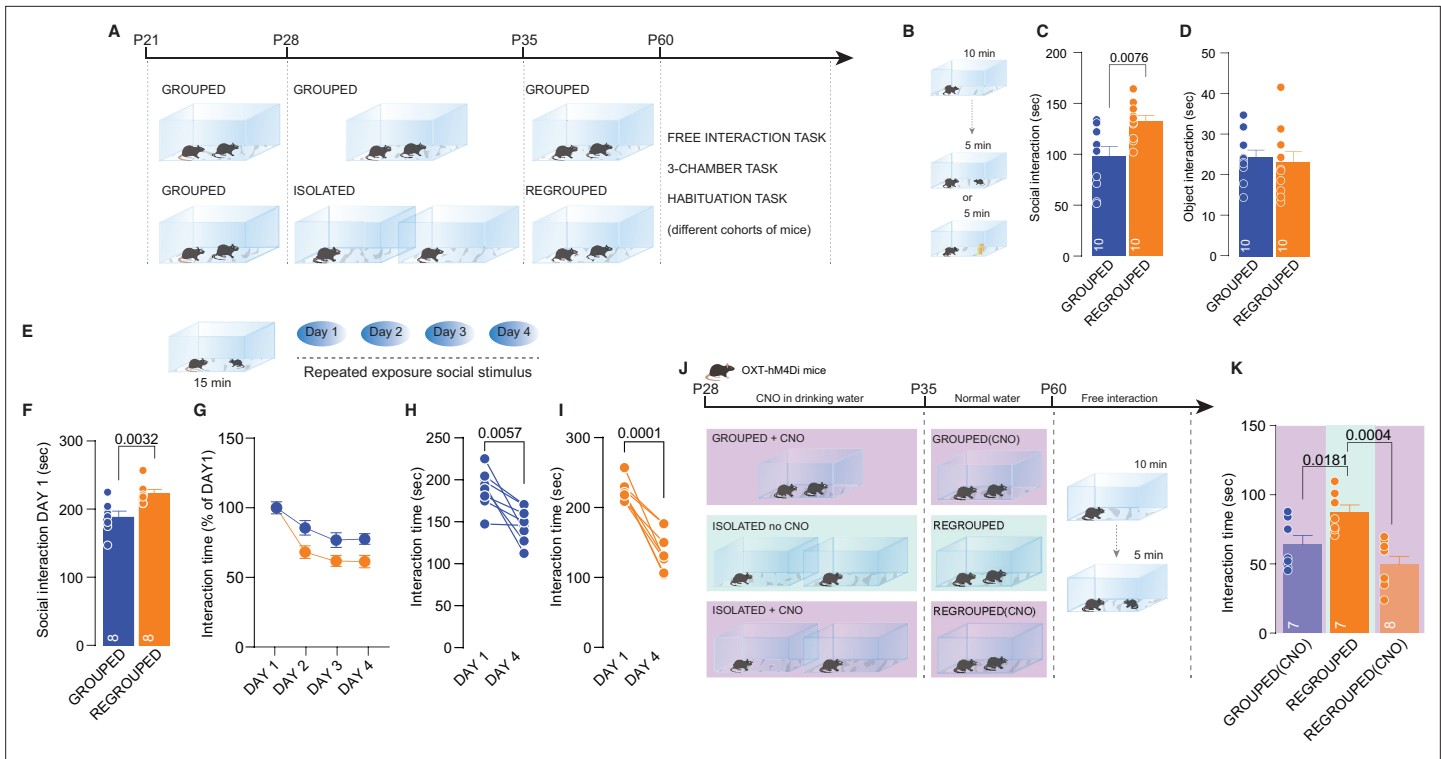

**Figure 4.** Long-lasting effects of adolescence acute social isolation. (**A**) Experimental design: WT mice were isolated between P28 and P35 and regrouped until P60 or always kept in group. Mice (different cohorts) were subjected to different behavioral tasks. (**B**) Free-interaction task paradigm. (**C**) Time exploring social stimulus (Unpaired-samples t-test, $t_{(18)}$=3.004, p=0.0076, n=10 mice each group). (**D**) Time exploring object (Unpaired-samples t-test, $t_{(18)}$=3.3717, p=0.7144, n=10 mice each group). (**E**) Habituation task paradigm. (**F**) Interaction time on Day 1 (Unpaired-samples t-test, $t_{(14)}$=3.553, p=0.0032, n=8 mice each group). (**G**) Interaction time across 4 days (Two-way RM ANOVA, Day × House condition $F_{(3, 42)}$=2.607, p=0.064, Day main effect $F_{(2.840, 39.76)}$=32.66, p<0.001, House condition main effect $F_{(1, 14)}$=8.240, p=0.0123, n=8 mice each group). (**H**) Interaction time during Day 1 and Day 4 for Grouped mice (Paired samples t-test, $t_{(7)}$=3.923, p=0.0057). (**I**) Interaction time during Day 1 and Day 4 for Regrouped mice (Paired samples t-test, $t_{(7)}$=7.621, p<0.001). (**J**) Experimental design. Oxt-hM4Di mice were isolated from P28 to P35 and regrouped until P60 or kept always grouped. CNO was dissolved in drinking water and administered during social isolation. At P60 mice underwent to free direct social interaction task. (**K**) Social interaction time (One-way ANOVA followed by Tukey's multiple comparisons test, $F_{(2, 19)}$=9.430, p=0.0014, Grouped (CNO) n=7, Regrouped n=7, Regrouped (CNO) n=8). Data are represented as mean ± SEM.

The online version of this article includes the following source data and figure supplement(s) for figure 4:

**Source data 1.** *Figure 4* - raw data and statistical output.

**Figure supplement 1.** Long-lasting effects of acute social isolation in adolescence.

**Figure supplement 1—source data 1.** *Figure 4—figure supplement 1* - raw data and statistical output.

**Figure supplement 2.** Regrouping after adolescence social isolation restore PVN and VTA excitability.

**Figure supplement 2—source data 1.** *Figure 4—figure supplement 2* - raw data and statistical output.

neuron-dependent changes in synaptic transmission at excitatory inputs onto pDA neurons projecting to the PFC.

Finally, to assess the causal link between the presence of CP-AMPARs and isolation-dependent changes in social interaction, the VTA in each mouse was cannulated, and we injected a CP-AMPAR antagonist (NASPM) into the region 10 min before the direct interaction task (***Figure 6A and B***). While NASPM did not affect the interaction time in control mice, the inhibition of CP-AMPARs in the VTA was sufficient to normalize social interaction in the regrouped mice (***Figure 6C***). The data presented here show that the increased activity of oxytocin neurons during social interaction induces synaptic scaling exerted by the presence of CP-AMPARs and that these receptors are responsible for increased social interaction during adulthood.

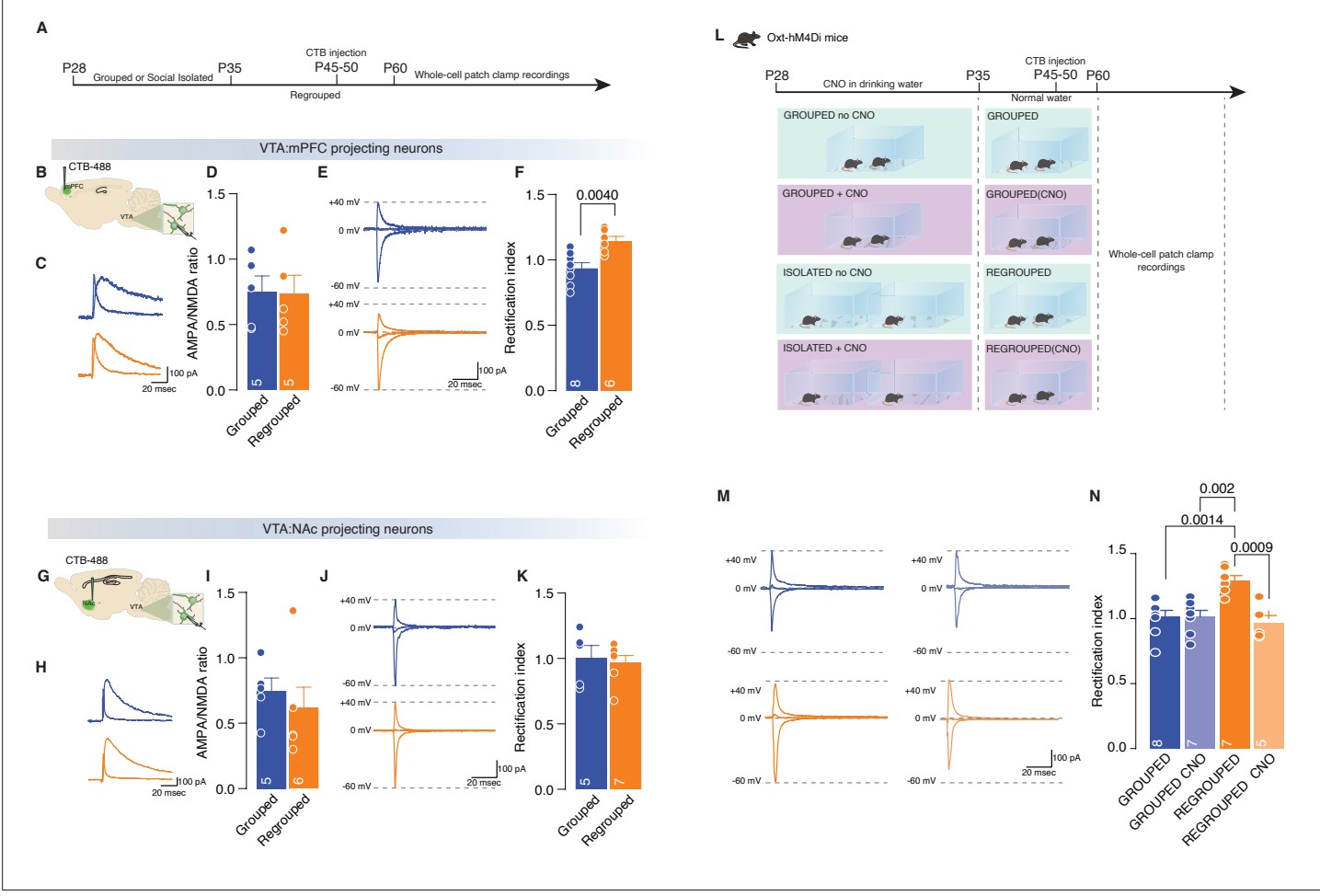

**Figure 5.** Adolescence acute social isolation induces synaptic scaling in adulthood mice. (**A, B, G**) Experimental paradigm. WT mice were isolated between P28 and P35. Then mice were regrouped, injected with 488-CTB in the mPFC (**B**) or NAc (**G**) between P45–P50 and at P60 were subjected at whole-cell patch-clamp recording. (**C**) Example traces of isolated AMPA and NMDA currents recorded at +40 mV. (**D**) AMPA-NMDA ratio of VTA-DA:mPFC projecting neurons (Unpaired samples t-test, $t_{(8)}$=0.07544, p=0.9417, Grouped n=5, Isolated n=5 from two mice each group). (**E**) Example traces of Isolated AMPA current recorded at +40, 0, and –60 mV. (**F**) Rectification index (RI) of VTA-pDA:mPFC projecting neurons (Unpaired samples t-test, $t_{(12)}$=3.545, p=0.004, Grouped n=8, Isolated n=8 from two mice each group). (**H**) Example traces of isolated AMPA and NMDA currents recorded at +40 mV. (**I**) AMPA-NMDA ratio of VTA-pDA:NAc projecting neurons (Mann-Whitney U-test=7, p=0.1602, Grouped n=5, Isolated n=6 from two mice each group). (**J**) Example traces of Isolated AMPA current recorded at +40, 0, and –60 mV. (**K**) RI of VTA-pDA:NAc projecting neurons (Unpaired samples t-test, $t_{(10)}$=0.3720, p=0.7176, Grouped n=5, Isolated n=7 from two mice each group). (**L**). Experimental paradigm: Oxt-hM4Di mice were isolated between P28 and P35 or kept always grouped and CNO was dissolved in drinking water. Then mice were regrouped, injected with CTB at P45–P50, and kept group-housed until P60 when were subjected at whole-cell patch-clamp recording. (**M**) Example traces of isolated AMPA current recorded at +40, 0, and –60 mV. (**N**) RI of VTA-pDA:mPFC projecting neurons (Two-way ANOVA followed by Tukey's multiple comparisons test, CNO treatment main effect $F_{(1, 23)}$=11.19, p=0.0028, House condition main effect $F_{(1, 23)}$=5.459, p=0.0285, CNO treatment × Hous condition $F_{(1, 23)}$=11.22, p=0.028, Grouped n=8, Grouped CNO n=7, Regrouped n=7, Regrouped CNO n=5 from two mice each group). Data are represented as mean ± SEM.

The online version of this article includes the following source data and figure supplement(s) for figure 5:

**Source data 1.** *Figure 5* - raw data and statistical output.

**Figure supplement 1.** Adolescence acute social isolation induces synaptic scaling in VTA pDA-mPFC projecting neruons.

**Figure supplement 1—source data 1.** *Figure 5—figure supplement 1* - raw data and statistical output.

**Figure supplement 2.** Immunohistochemical validation of VTA-DA neurons.

## Discussion

Social isolation is an adverse experience across social species with long-lasting behavioral and physiological consequences (*Cacioppo et al., 2014*). Although anxiety, depression, and obsessive-compulsive behavior generally emerge after prolonged social isolation (*Teo et al., 2013*; *Pietrabissa*

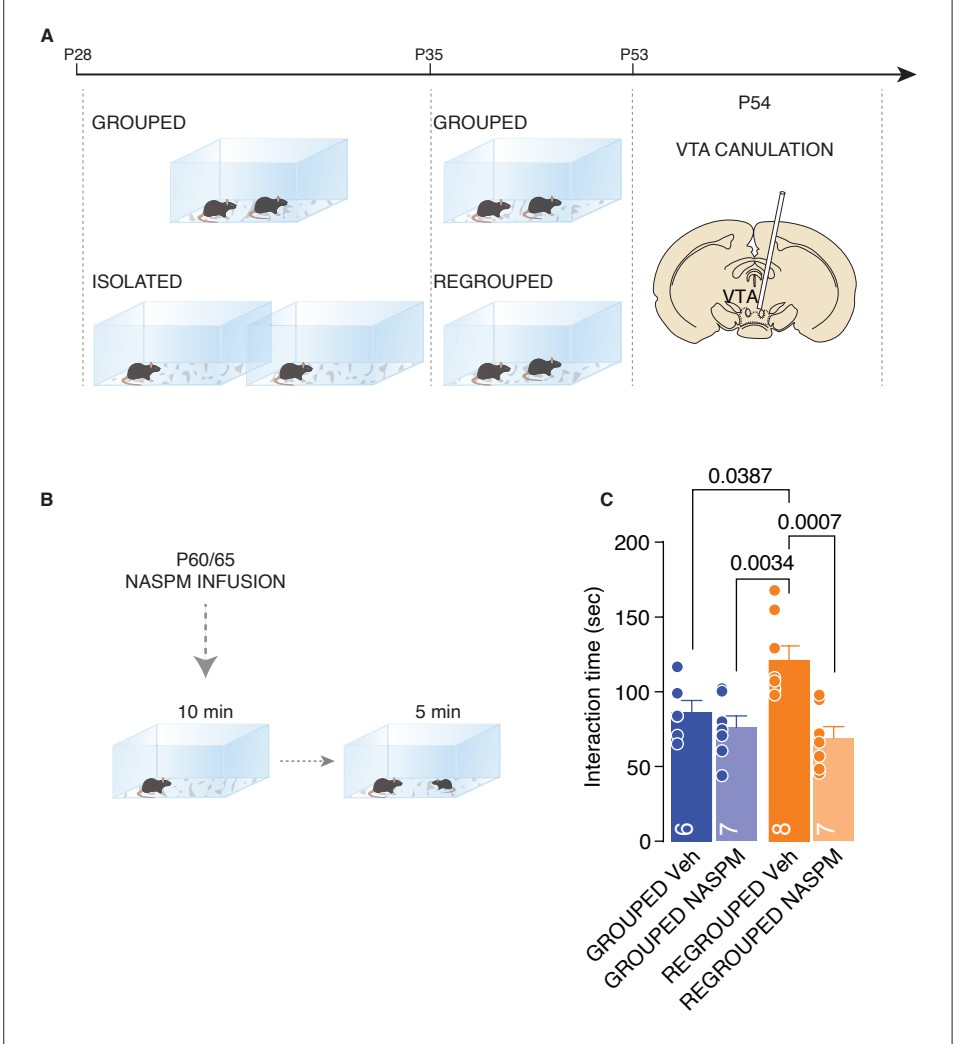

**Figure 6.** CP-AMPARs are responsible of increased social interaction during adulthood. (**A**) Experimental paradigm: WT mice were isolated between P28 and P35. Then mice were regrouped until P53 and canulated over the VTA. (**B**) Mice underwent to direct free interaction task after infusion of CP-AMPARs antagonist NASPM. (**C**) Social interaction time (Two-way ANOVA followed by Tukey's multiple comparisons test, NASPM treatment main effect $F_{(1, 24)}$=13.78, p=0.0011, House condition main effect $F_{(1, 24)}$=2.626, p=0.1182, NASPM × House condition $F_{(1, 24)}$=6.099, p=0.021, Grouped Veh n=6, Grouped NASPM n=7, Regrouped Veh n=8, Regropued NASPM n=7). Data are represented as mean ± SEM.

The online version of this article includes the following source data for figure 6:

**Source data 1.** *Figure 6* - raw data and statistical output.

---

*and Simpson, 2020*; *Leigh-Hunt et al., 2017*), it is evident that a short period of deprivation from conspecific interaction may also trigger aversive consequences. The adverse effects of social isolation are particularly apparent for adolescents (*Almeida et al., 2021*; *Orben et al., 2020*). Indeed, adolescence is a critical period for not only social interaction (*Larsen and Luna, 2018*; ) but also the emergence of psychiatric diseases (*Orben et al., 2020*; *Marco et al., 2011*), and social isolation during adolescence can exacerbate mental health problems (*Larsen and Luna, 2018*; *Almeida et al., 2021*; ; *Graf et al., 2021*; *Andersen, 2021*). Why is adolescence significantly affected, and what are the circuit mechanisms underlying the negative consequences of social isolation?

Social interaction is a basic need across species, and rodents are excellent models for investigating the neural bases of social interaction and social isolation. During adolescence, social and cognitive skills depend on peer-to-peer interactions (*Vanderschuren et al., 1997*; *Baarendse et al., 2013*), and acute social isolation in rodents is sufficient to cause behavioral abnormalities, including increased

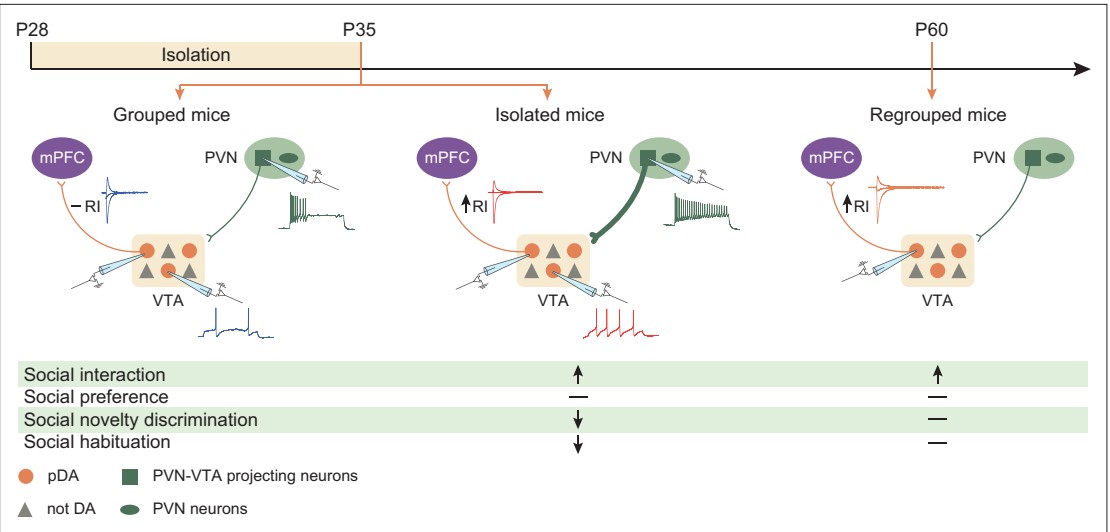

**Figure 7.** Graphical summary. One week of social isolation during adolescence in mice increased social interaction at the expense of social habituation and social novelty preference. Behavioral changes were accompanied by the acute hyperexcitability of PVN-VTA projecting neurons, VTA putative dopamine (pDA) neuron, and long-lasting expression of GluA2-lacking AMPARs at excitatory inputs onto pDA neurons that project to the prefrontal cortex (PFC).

vulnerability to drug addiction and depressive-like behavior, later in life (*Lampert et al., 2017*; *Takatsu-Coleman et al., 2013*; *Whitaker et al., 2013*). Here, we show that 1 week of social isolation during adolescence is sufficient to generate a rebound increase in social interaction during the free-interaction task in male mice, independently whether the stimulus is a former cage mate or a novel stimulus. The increase in interaction is accompanied by deficits in social novelty preference and social habituation across days. However, social isolation does not affect short social habituation, suggesting that isolation does not affect short-term social memory (*Figure 7*). These data support the hypothesis that social interaction is a need and that the absence of social cues generates a craving response like that caused by food cues after fasting (*Tomova et al., 2020*). Indeed, we could speculate that social isolation during adolescence results in an intense urge to interact with whatever conspecific is present.

Previous studies have indicated the role of DA neurons in the VTA in social behavior (*Gunaydin et al., 2014*; *Bariselli et al., 2018*; *Bariselli et al., 2016b*) and shown that chronic isolation alters DA levels in the NAc (*Hall et al., 1998*; ). Remarkably, acute social isolation during adulthood changes the synaptic strength at excitatory inputs onto DA neurons of the DRN in rodents, leaving synaptic transmission onto DA neurons of the VTA unaltered (*Matthews et al., 2016*). It has been shown that human exposure to social cues after acute isolation evokes activity in the VTA (*Tomova et al., 2020*), suggesting that while DA neurons within the DRN mediate the loneliness-like brain state induced by social isolation, DA neurons in the VTA mediate social craving. Our data indicate that 1 week of isolation during adolescence is sufficient to increase the overall excitability of neurons within the meso-corticolimbic system. We next investigated the underlying mechanisms and found that the activity of oxytocin neurons within the PVN is causally linked to neuronal excitability within the reward system and the behavioral consequences of social isolation. Previously, the release of oxytocin in the VTA was shown to increase DA neuron firing within the VTA (*Xiao et al., 2017*).

Furthermore, restoration of oxytocin signaling in the VTA in an autism spectrum disorder-related mouse model was sufficient to rescue social novelty responses (*Hörnberg et al., 2020*). Taken together, these data suggest that oxytocin neurons could represent the neural signature of social craving in male mice. Although sex differences in oxytocin and oxytocin receptor distribution have not been reported, sex differences in oxytocin's behavioral effects can be found in rodents (*Miller and Caldwell, 2015*; *Tamborski et al., 2016*; *Hammock and Levitt, 2013*). Indeed, neonatal oxytocin treatment modifies the development of social behavior resulting in a reduction of alloparenting (*Mogi et al., 2014*) and treatment during puberty reduces social play behavior (*Bredewold et al., 2014*). Both these effects occur only in female. The lack of female in the present study poses a limit in our results since we do not know whether our results apply to both sexes. In the brain there are

sex differences in the tissue structures and the dimorphic nuclei exhibiting morphological sex differences are considered as the structural basis for generating sex differences in brain functions (*Ogawa et al., 2020*). Gonadal steroids regulate Oxt activity, and among them, estrogen seems to have higher impact on the Oxt system (*Richard and Zingg, 1990*; *Miller and Caldwell, 2015*; *Gimpl and Fahrenholz, 2001*). Indeed, estrogens play a fundamental role in the sexually dimorphic formation of brain structures by modulating neural circuits that control sex-specific behavior, including social interaction (*Ogawa et al., 2020*). Future studies will aim to study the effects of juvenile social isolation in females and investigate the role of the oxytocin system in isolation-dependent changes in social behavior.

We next chose to investigate whether acute social isolation during adolescence has long-lasting consequences. Isolated mice spent more time interacting with a conspecific than control mice did 1 month after regrouping, indicating that the rebound increase in free/unrestrained social interaction in a long-lasting effect of isolation . Recently, it was reported that 2 weeks of social isolation after weaning altered the pathway from the medial PFC to the posterior paraventricular thalamus, leading to decreased sociability in adult mice (*Yamamuro et al., 2020*). Although the duration of social isolation can explain the differences at the behavioral level between the two studies, these data show that adolescence is a critical period for the establishment of social behavior and that isolation during this period alters sociability during adulthood.

During adulthood, increased social interaction is accompanied and causally linked to the increase in GluA2-lacking AMPARs at excitatory inputs onto pDA neurons projecting to the PFC. GluA2-lacking AMPARs are CP ionotropic receptors with larger single-channel conductance and can undergo voltage-dependent blockade by polyamines (*Twomey et al., 2018*; *Man, 2011*). These receptors are inserted at excitatory synapses after exposure to addictive drugs (*Bellone and Lüscher, 2006*) and have been proposed to mediate the incubation of drug craving (*Conrad et al., 2008*). Moreover, it has also been shown that stress induces the insertion of these receptors, and local blockade of GluA2-lacking AMPARs was shown to attenuate stress-induced behavioral changes in rodents (*Kuniishi et al., 2020*; *Yi et al., 2017*). Furthermore, the presence of these receptors changes the rules for the induction of plasticity (*Mameli et al., 2011*) and mediates in vivo synaptic scaling (*Garcia-Bereguiain et al., 2013*). Here, we show that prolonged activation of DA neurons induced by social isolation promoted the insertion of CP-AMPARs. In vivo blockade of these receptors was sufficient to rescue social isolation-induced behavioral deficits. Taken together, all these findings indicate that GluA2-lacking AMPARs are critical therapeutic targets for treating maladaptive motivated behavior and suggest that inhibitors of these receptors may counteract the negative consequences of social isolation.

By investigating the short-term and long-term consequences of social isolation during adolescence, our work contributes to our understanding of how social isolation impacts neural circuits and behavior. Since social isolation has a tremendous impact on mental health and increases vulnerability to psychiatric diseases, our work is relevant to identifying new therapeutic approaches to ameliorate maladaptive social behavior.

## Materials and methods

**Key resources table**

| Reagent type (species) or resource | Designation | Source or reference | Identifiers | Additional information |
|---|---|---|---|---|
| Chemical compound, drug | Choline chloride | Sigma-Aldrich | C7527 | |
| Chemical compound, drug | D-(+)-Glucose | Sigma-Aldrich | G8270 | |
| Chemical compound, drug | D(+)-Saccharose | Roth | 4621.1 | |
| Chemical compound, drug | NaHCO$_3$ | Sigma-Aldrich | S5761 | |
| Chemical compound, drug | MgCl$_2$ | Sigma-Aldrich | 63068 | |
| Chemical compound, drug | Ascorbic acid | Roth | 6288.1 | |
| Chemical compound, drug | Sodium pyruvate | Roth | 8793.1 | |
| Chemical compound, drug | KCl | Sigma-Aldrich | 60130 | |
| Chemical compound, drug | NaH$_2$PO$_4$ | Sigma-Aldrich | S0751 | |

*Continued on next page*

*Continued*

| Reagent type (species) or resource | Designation | Source or reference | Identifiers | Additional information |
|---|---|---|---|---|
| Chemical compound, drug | CaCl2 | Sigma-Aldrich | 21097 | |
| Chemical compound, drug | NaCl | Sigma-Aldrich | 31434 | |
| Chemical compound, drug | K-Gluconate | Sigma-Aldrich | G4500 | |
| Chemical compound, drug | EGTA | Sigma-Aldrich | E4378 | |
| Chemical compound, drug | HEPES | Sigma-Aldrich | H3375 | |
| Chemical compound, drug | Na2ATP | Sigma-Aldrich | A2383 | |
| Chemical compound, drug | Na3GTP | Sigma-Aldrich | G8877 | |
| Chemical compound, drug | Creatine-phosphate | Sigma-Aldrich | P7936 | |
| Chemical compound, drug | CsCl | Sigma-Aldrich | 20966 | |
| Chemical compound, drug | Sodium creatine phosphate | Sigma-Aldrich | 27920 | |
| Chemical compound, drug | Spermine | Sigma-Aldrich | S4264 | |
| Chemical compound, drug | Lidocaine N-ethyl bromide (QX-314) | Sigma-Aldrich | L5783 | |
| Chemical compound, drug | NaOH | Sigma-Aldrich | S5881 | |
| Chemical compound, drug | Clozapine N-oxide | Enzo Life | BML-NS-105-0025 | 5 mg/200 ml |
| Chemical compound, drug | Saccharin | Sigma-Aldrich | 240931 | |
| Chemical compound, drug | NASPM | Tocris | 2766 | |
| Chemical compound, drug | NGS | Sigma-Aldrich | AB7481 | |
| Chemical compound, drug | Triton X-100 | Sigma-Aldrich | X100 | |
| Chemical compound, drug | Kynurenic acid | Sigma-Aldrich | K3375 | |
| Chemical compound, drug | Picrotoxin | Tocris | 1128 | |
| Chemical compound, drug | D-APV | Tocris | 106 | |
| Antibody | Oxytocin antibody (guinea pig polyclonal) | Synaptic Systems | #408004 | (1:500) |
| Antibody | cFOS antibody (rabbit monoclonal) | Synaptic Systems | #226008 | (1:5000) |
| Antibody | Neurophysin 2/NP-AVP antibody (mouse monoclonal) | Merck Millipore | #MABN856 | (1:250) |
| Antibody | HA-tag (6E2) antibody (mouse monoclonal) | Cell Signaling Technology | #2367 | (1:200) |
| Antibody | Oxytocin antibody (rabbit polyclonal) | Merck Millipore | #AB911 | (1:500) |
| Antibody | Tyrosine Hydroxylase antibody (mouse monoclonal) | Merck Millipore | #MAB318 | (1:500) |
| Antibody | Goat anti-mouse Alexa Fluor 555 (polyclonal) | Thermo Fisher Scientific | A21422 | (1:500) |
| Antibody | Goat anti-rabbit Alexa Fluor 488 (polyclonal) | Thermo Fisher Scientific | A11008 | (1:500) |
| Antibody | Goat anti-mouse Alexa Fluor 594 (polyclonal) | Thermo Fisher Scientific | A11005 | (1:500) |
| Antibody | Goat anti-mouse Alexa Fluor 647 (polyclonal) | Thermo Fisher Scientific | A21235 | (1:500) |
| Antibody | Goat anti-guinea pig Alexa Fluor 555 (polyclonal) | Thermo Fisher Scientific | A21435 | (1:500) |
| Other | Mounting medium with DAPI | Abcam | ab104139 | Mounting media used for immunofluorescence |
| Other | Alexa Fluor-488 hydrazydes | Thermo Fisher Scientific | #A10436 | 10 mM—GFP tracer used for DA neurons validation (*Figure 5—figure supplement 2*) |
| Other | Cholera Toxin Subunit B, Alexa Fluor 488 Conjugate | Thermo Fisher Scientific | #C34775 | Retrograde tracer used in *Figures 3 and 5* and *Figure 5—figure supplement 1* |

## Animals and experimental paradigms

The study was conducted with male wild-type (WT), and R26-hM4Di/mCitrine mice (B6.129-Gt(ROSA)26Sor[tm1(CAG-CHRM4*,-mCitrine)Ute]/J, Jackson stock #026219) crossed with Oxytocin-Ires-Cre mice (B6;129S-Oxt[tm1.1(cre)Dolsn]/J, Jackson stock #024234) allowing thus the expression of inhibitory DREADD under *Oxytocin* promoter (Oxt-hM4Di mice), under 12 hr light-dark cycle (7 a.m. to 7 p.m.) with free access to food and drink. Behavioral experiments were conducted in a room with fixed low illumination (10–15 lux) and with controlled humidity (40%) and temperature (22–24°C). The experiments were always performed within a time frame that started approximately 1.5 hr after the end of the dark cycle and ended 2 hr before the start of the next dark cycle. For adolescence social isolation, mice were weaned at P21 and kept in groups until P28. Subsequently, mice were single used until P35. For 24-hr isolation, mice were single used between P34 and P35. For social isolation in adulthood, mice were kept in groups until P53 and single-housed until P60. For the long-lasting effect of SI, mice were singly housed between P28 and P35 and subsequently regrouped until P60. When group-housed, mice were housed two per cage. The experimental mice were randomly assigned to the different groups. All the procedures performed at UNIGE complied with the Swiss National Institutional Guidelines on Animal Experimentation and were approved by the Swiss Cantonal Veterinary Office Committees for Animal Experimentation.

## Direct free interaction task

Mice were allowed to freely explore the arena for 10 min (clean homecage, 35×20 cm$^2$ with transparent wall), and subsequently an unfamiliar sex-matched conspecific social stimulus (always 1 week younger to promote social play behavior) or an object was introduced, and the interaction time was recorded for 5 min. For the social interaction with a former cage mate, the arena was divided into two parts with a wall, and the experimental mouse was put on one side while the stimulus was added on the other side. After 10 min, the wall was removed, and the mice were free to interact. We used this modified protocol to reduce the number of used mice because in this way, we could have two interaction score values simultaneously for the grouped mice. All the trials were recorded with a camera placed above the arena. The interaction time was manually scored when the experimental mouse initiated the action and when the nose of the animal was oriented toward the social stimulus mouse only or toward the object.

## Three-chamber task

A three-chambered social interaction assay was used, comprising a rectangular Plexiglas arena (60×40×22 cm$^3$ with opaque walls) (Ugo Basile, Varese, Italy) divided into three chambers (each 20×40×22 (h) cm$^3$). Each mouse was placed in the arena for a habituation period of 10 min when it was allowed to explore the empty arena freely. At the end of the habituation, two enclosures with vertical metal bars were placed in the center of the two outer chambers. One enclosure was empty (serving as an object), whereas the other contained a social stimulus (stimulus 1, 1 week younger unfamiliar sex-matched conspecific). The experimental mouse was allowed to freely explore the apparatus and the enclosures for 10 min (social preference phase). Subsequently, the empty enclosure was replaced with another containing an unfamiliar conspecific social stimulus (stimulus 2). The experimental mouse was allowed to freely explore the apparatus for 10 min and the enclosures for another 10 min (social novelty phase). The position of the empty versus social stimulus 1-containing or social stimulus 1-containing versus social stimulus 2-containing enclosures alternated and was counterbalanced for each trial to avoid any bias effects. Every session was video-tracked and recorded using ANY-maze (Stoelting Europe, Dublin, Ireland), which provided an automated recording of the time in the compartment, and the distance moved. The time spent interacting with each enclosure was manually scored and then used to determine the preference index for the object or social target (stimulus 1 and stimulus 2). The stimulus interaction was scored when the nose of the experimental mouse was oriented toward the enclosures at a distance of approximately less than 2 cm. The arena was cleaned with 1% acetic acid solution and dried between trials. For the analysis, we calculated the 'Preference index' using the following formula: interaction time target 1/(interaction time target 1+interaction time target 2) or target 2/(interaction time target 1+interaction time target 2). Using this formula, the threshold is 0.5, which corresponds to the chance level to explore either target.

## Long and short habituation task

A clean homecage was used as an arena (35×20 cm$^2$ with transparent wall). For the long habituation task protocol, the experimental mouse was placed in the arena with a novel social stimulus (sex-matched conspecific mouse, 1 week younger compared to the experimental mouse). The animals were left freely to explore the cage and interact with each other for 15 min. At the end of the trial, the experimental and stimulus mice were returned to their homecage. For 4 consecutive days, the experimental mouse was exposed to the same social stimulus. For the short habituation task protocol, the experimental mouse was placed in the arena for 15 min. After a novel social stimulus (sex-matched conspecific mouse, 1 week younger compared to the experimental mouse) was introduced. The animals were left freely to interact with each other for 2 min. At the end of the trial, the stimulus mice were returned to their homecage, while the experimental mouse stayed in the arena alone for 5 min. At the end of the inter-trial interval, the same stimulus mouse was returned in the arena for another 2 min. The exposure was repeated four times in a row. All the trials were recorded with a camera placed above the arena. Non-aggressive behavior was manually scored when the experimental mouse initiated the action and when the nose of the animal was oriented toward the social stimulus mouse only.

## Novel object recognition

A squared arena was used for the task, consisting of three phases: a first habituation phase followed by a familiarization and the actual test phase. During the habituation phase, the experimental mouse is let to freely explore the arena (40×40×40 cm$^3$ with dark wall) for 10 min. During the familiarization phase, the animal was exposed to two identical objects (object 1 – object 2) and was let to interact for 10 min with both freely. After a retention delay of 20 min, the mice were exposed for 10 min to one of the familiar objects (object 1), while the other was replaced with a novel object (object 3). During the different phases of the test, the objects were placed on the opposite sides of the cage, alternating the position of the respective objects. Every session was video-tracked and recorded using ANY-maze (Stoelting Europe, Dublin, Ireland). The time spent interacting with each object was manually scored and then used to determine the preference index for the different objects. The stimulus interaction was scored when the nose of the experimental mouse was oriented toward the objects at a distance of approximately less than 2 cm. The arena was cleaned with 1% acetic acid solution and dried between trials. For the analysis, we calculated the 'Preference index' using the following formula: interaction time target 1/(interaction time target 1+interaction time target 2) or target 2/(interaction time target 1+interaction time target 2). Using this formula, the threshold is 0.5, which corresponds to the chance level to explore either one target.

## Elevated plus maze

The elevated plus maze (EPM) consisted of a platform of four opposite arms (40 cm), two of them are open, and the other two are closed (enclosed by 15 cm high walls). The apparatus was elevated at 55 cm from the floor. Each male adult mouse was placed individually in the center of the EPM apparatus with the snout facing one of the open arms and was filmed for 5 min. Distance moved (cm) and time spent in the open and closed arms (s) of the arena were measured with ANY-maze (Stoelting Europe, Dublin, Ireland). The apparatus was cleaned with 1% acetic acid solution and dried between trials between each session. For the analysis, we calculated the 'Preference index' using the following formula: interaction time target 1/(interaction time target 1+interaction time target 2) or target 2/(interaction time target 1+interaction time target 2). Using this formula, the threshold is 0.5, which corresponds to the chance level to explore either one target.

## Social conditioned place preference

The apparatus consists of two square-shaped chambers (20×20 cm$^2$) with either gray stripes on white background or black dots on white background. The floor in each of the two chambers has different distinct textures. The two chambers are interconnected by a small corridor, with transparent walls and floor. The task is divided into three phases: Day 1, 15 min pre-TEST in which the experimental mouse is free to explore the entire apparatus; Days 2–5 conditioning phase (30 min per day for 4 consecutive days); Day 6, 15 min post-TEST, in which the experimental mouse is left freely to explore the apparatus in absence of stimuli. For the conditioning phase, one chamber was randomly assigned as

the paired session chamber, with the presence of an unfamiliar conspecific mouse (social chamber) and the other as the non-social session chamber (non-social chamber). During the conditioning, for each day, the experimental mouse was left in the social chamber or in the non-social chamber for 5 min each session, alternating for six times in the two chambers (30 min of conditioning). During each conditioning session, the experimental mice were allowed to freely interact with social stimuli that remained the same one for the entire conditioning phase (typically, one stimulus mouse was assigned to one experimental animal). The behavior of the animals was tracked automatically with the ANY-maze software (Stoelting Europe, Dublin, Ireland), and the time spent in the two chambers was recorded for the pre- and post-TEST sessions. Preference index for social chamber was calculated as: time spent in the social chamber divided by the total time spent in the two chambers.

## Whole-cell patch-clamp recordings

Horizontal midbrain slices 200-µm thick containing the VTA or coronal midbrain slices 250-µm thick containing PVN were prepared. Brains were sliced by using a cutting solution containing: 90.89 mM choline chloride, 24.98 mM glucose, 25 mM $NaHCO_3$, 6.98 mM $MgCl_2$, 11.85 mM ascorbic acid, 3.09 mM sodium pyruvate, 2.49 mM KCl, 1.25 mM $NaH_2PO_4$, and 0.50 mM $CaCl_2$. Brain slices were incubated in cutting solution for 20–30 min at 35°. Subsequently, slices were transferred in artificial cerebrospinal fluid (aCSF) containing: 119 mM NaCl, 2.5 mM KCl, 1.3 mM $MgCl_2$, 2.5 mM $CaCl_2$, 1.0 mM $NaH_2PO_4$, 26.2 mM $NaHCO_3$, and 11 mM glucose, bubbled with 95% $O_2$ and 5% $CO_2$ at room temperature. Whole-cell voltage-clamp or current-clamp electrophysiological recordings were conducted at 35–37° in aCSF (2–3 ml/min, submerged slices). Recording pipette contained the following internal solution: 140 mM K-Gluconate, 2 mM $MgCl_2$, 5 mM KCl, 0.2 mM EGTA, 10 mM HEPES, 4 mM $Na_2ATP$, 0.3 mM $Na_3GTP$, and 10 mM creatine-phosphate. The cells were recorded at the access resistance from 10 to 30 MΩ. Resting membrane potential (in mV) was read using the Multiclamp 700B Commander (Molecular Devices) while injecting no current (I=0) immediately after breaking into a cell. Action potentials (APs) were elicited in current-clamp configuration by injecting depolarizing current steps (50 pA, 500 ms) from 0 to 400 pA in presence.

For VTA excitability, pDA neurons were identified accordingly to their position (medially to the medial terminal nucleus of the accessory optic tract), morphology (large soma), and cell capacitance (> 28 pF, see *Figure 5—figure supplement 2*). For CNO validation (20 µM), the drug was applied in the recording chamber before starting the excitability protocol, and oxytocin cells were recognized, exciting the tissue with 594 nm LED, allowing thus the visualization of the mCitrine fluorescence. Excitatory postsynaptic currents (EPSCs) were recorded in voltage-clamp configuration, elicited by placing a bipolar electrode rostro-laterally to VTA at 0.1 Hz and isolated by applying the $GABA_AR$ antagonist picrotoxin (100 µM). Recording pipette contained the following internal solution: 130 mM CsCl, 4 mM NaCl, 2 mM $MgCl_2$, 1.1 mM EGTA, 5 mM HEPES, 2 mM $Na_2ATP$, 5 mM sodium creatine phosphate, 0.6 mM $Na_3GTP$, 0.1 mM spermine, and 5 mM lidocaine N-ethyl bromide (QX-314). Access resistance (10–30 MΩ) was monitored by a hyperpolarizing step of –4 mV at each sweep, every 10 s. Data were excluded when the resistance changed >20%. The AMPA/NMDA was calculated by subtracting the mixed EPSC (+40 mV), the non-NMDA component isolated by D-APV (50 µM at +40 mV) bath application. The ratio values may be underestimated since it was calculated with spermine in the pipette. The RI of AMPARs is the ratio of the chord conductance calculated at a negative potential (–60 mV) divided by the chord conductance at positive potential (+40 mV). For sIPSCs were recorded from VTA pDA neurons in the presence of 3 mM kynurenic acid. The patch pipettes were filled with (in mM): 30 K-gluconate, 100 KCl, 10 creatin-phosphate, $MgCl_2$, 3.4 $Na_2ATP$, 0.2 $Na_3GTP$, 1.1 EGTA, 5 HEPES, pH adjusted to 7.3 with NaOH, osmolarity to 289 mOsm. All the synaptic responses were collected with a Multiclamp 700B-amplifier (Axon Instruments, Foster City, CA), filtered at 2.2 kHz, digitized at 5 kHz, and analyzed online using Igor Pro software (Wavemetrics, Lake Oswego, OR).

## Surgeries

Injections of Cholera-toxin subunit B (CTB)-Alexa Fluor 488 (Thermo Fisher Scientific #C34775) conjugated were performed in WT and Oxt-hM4Di mice at P21 or P45–P50. Mice were anesthetized with a mixture of oxygen (1 L/min) and isoflurane 3% (Baxter AG, Vienna, Austria) and placed in a stereotactic frame (Angle One, Leica, Germany). The skin was shaved, locally anesthetized with 40–50 µl lidocaine 0.5%, and disinfected. Unilateral or bilateral craniotomy (1 mm in diameter) was then performed at

following stereotaxic coordinates: NAc ML ±0.85 mm, AP +1.3 mm, DV –4.5 mm from Bregma; mPFC (four injection site) position 1 ML ±0.27 mm, AP +1.5 mm, DV –3, –25 mm, position 2 ML ±0.27 mm, AP +1.75 mm, DV –3, –2.5 mm, position 3 ML ±0.27 mm, AP +2 mm, DV –2.6, –2.2 mm, position 4 ML ±0.27 mm, AP +2.25 mm, DV –2 mm from Bregma; VTA AP –3, ML ±0.5, DV –4.3 from bregma. The CTB was injected via a glass micropipette (Drummond Scientific Company, Broomall, PA) either into the NAc, mPFC, and VTA at the rate of 100 nl/min for a total volume of 200 nl on each side. For NASPM experiments, unilateral implantations of stainless steel 26-gauge cannula (Plastics One, VI) were performed on WT mice at P54. Mice were anesthetized and placed in a stereotactic frame as previously described. Unilateral craniotomy (1 mm in diameter) was performed over the VTA at the following stereotactic coordinates: ML: ± 0.9 mm, AP: –3.2 mm, DV: –3.95 mm from Bregma. The cannula was implanted at a 10° angle, placed above the VTA, and fixed on the skull with dental acrylic. A removable cap protected the cannula. All animals underwent behavioral experiments 1–2 weeks after surgery.

## Pharmacological treatments

Isolated or grouped mice were treated for 1 week with Clozapine N-oxide (CNO, Enzo Life Science, Farmingdale). CNO was dissolved in the drinking water at 5 mg/200 ml in 4% sucrose and 0.2% saccharin solution. Mice received either CNO solution or sugar solution only as control. The solutions were prepared fresh daily. For the acute effects of SI, after 1 week of treatment, mice underwent a direct free interaction task or were used for whole-cell patch-clamp recordings. For the long-lasting effect of SI, CNO treatment was stopped at P35, mice were regrouped until P60, and normal water was given. Mice underwent direct free interaction tasks or were used for whole-cell patch-clamp recordings. For the experiments with 1-Naphthylacetyl spermine trihydrochloride (NASPM), mice were infused using a Minipump injector (pump Elite 11, Harvard apparatus, USA). Ten minutes before each trial, mice were either infused with four µg of NASPM dissolved in 500 L of aCSF (2 min of active injection at 250 nl/min rate, and 1 min at rest) or aCSF only (vehicle). After infusion, mice underwent to direct free interaction task.

## Immunofluorescence and images acquisition

For the staining in *Figure 3*, PVN slices were washed three times with phosphate-buffered saline (PBS) at 0.1 M. Slices were then pre-incubated with PBS-NGS-TX buffer (5% NGS and 0.3% Triton X-100) for 90 min at room temperature. Subsequently, slices were incubated with primary antibody (guinea pig anti-Oxytocin, 1/500 dilution, Synaptic Systems #408004, rabbit anti-cFOS 1/5000 dilution, Synaptic Systems #226008, and mouse anti-Neurophysin 2/NP-AVP 1/250 dilution, Merck Millipore #MABN856) diluted in PBS-BSA-TX (3% NGS and 0.3% Triton X-100) overnight at 4°C in the dark. The following day, slices were washed three times with PBS 0.1 M and incubated for 90 min at room temperature with the secondary antibody (1/500 dilution, goat anti-rabbit Alexa Fluor 488, Thermo Fisher Scientific A11008, goat anti-guinea pig Alexa Fluor 555, Thermo Fisher Scientific A21435, goat anti-mouse Alexa Fluor 647, and Thermo Fisher Scientific A21235), diluted in PBS-NGS-TX (3% NGS and 0.3% Triton X-100). Slices were washed three times with PBS 0.1 M, and finally coverslips were mounted using fluoroshield mounting medium with DAPI (Abcam, ab104139).

For the staining in *Figure 3—figure supplement 1*, PVN slices were washed three times with PBS at 0.1 M. Slices were then pre-incubated with PBS-NGS-TX buffer (5% NGS and 0.3% Triton X-100) for 90 min at room temperature. Subsequently, slices were incubated with primary antibody (mouse anti-HA-tag, 1/200 dilution, Cell Signaling Technology #2367, rabbit anti-Oxytocin 1/500 dilution, and Merck Millipore #AB911) diluted in PBS-NGS-TX (3% NGS and 0.3% Triton X-100) overnight at 4°C in the dark. The following day, slices were washed three times with PBS 0.1 M and incubated for 90 min at room temperature with the secondary antibody (1/500 dilution, goat anti-rabbit Alexa Fluor 488, Thermo Fisher Scientific A11008, goat anti-mouse Alexa Fluor 594, and Thermo Fisher Scientific A11005), diluted in PBS-NGS-TX (3% NGS and 0.3% Triton X-100). Slices were washed three times with PBS 0.1 M, and finally coverslips were mounted using fluoroshield mounting medium with DAPI (Abcam, ab104139).

Tissue images of PVN were acquired using a confocal laser-scanning microscope LSM700 (Zeiss). Images were analyzed with ImageJ software. The region of interest was delimited around PVN, and the number of Oxytocin, cFOS, and AVP positive cells were counted manually for each slice using

separate color channels. Subsequently, channels merge images were created for Oxytocin-cFOS, Oxytocin-AVP, and AVP-cFOS, and double-positive cells were manually counted for each merged image.

## VTA-DA neurons post hoc validation

Horizontal midbrain slices 200-µm thick containing the VTA were prepared as described before. Whole-cell voltage-clamp electrophysiological recordings were conducted at 35–37° in aCSF (2–3 ml/min, submerged slices). Putative VTA-DA neurons were identified for larger soma compared to the other cells and a capacitance >28 pF. Cells with capacitance lower than 27 pF were not considered putative-DA neurons (*Figure 5—figure supplement 2*). Recording pipette contained the following internal solution: 140 mM K-Gluconate, 2 mM MgCl2, 5 mM KCl, 0.2 mM EGTA, 10 mM HEPES, 4 mM Na2ATP, 0.3 mM Na3GTP and 10 mM creatine-phosphate, and 10 mM Alexa Fluor-488 hydrazides (Thermo Fisher Scientific, #A10436). Neurons were patched and filled with a GFP tracer for 10 min. Subsequently, the slice was fixed for 24 hr in PFA 4%. The slice was washed three times with PBS at 0.1 M and then pre-incubated with PBS-NGS-TX buffer (5% NGS and 0.3% Triton X-100) for 90 min at room temperature. Subsequently, the slice was incubated with primary antibody (mouse anti-TH, 1/500 dilution, Merck Millipore #MAB318) diluted in PBS-BSA-TX (3% NGS and 0.3% Triton X-100) overnight at 4°C in the dark. The following day, the slice was washed three times with PBS at 0.1 M and incubated for 90 min at room temperature with the secondary antibody (1/500 dilution, goat anti-mouse Alexa Fluor 555, and Thermo Fisher Scientific A21422) diluted in PBS-NGS-TX (3% NGS and 0.3% Triton X-100). Slices were washed three times with PBS at 0.1 M, and finally coverslips were mounted using fluoroshield mounting medium with DAPI (Abcam, ab104139). Tissue images of PVN were acquired using a confocal laser-scanning microscope LSM700 (Zeiss).

## Statistical analysis

Statistical analysis was conducted with GraphPad Prism 9 (San Diego, CA). Statistical outliers were identified with the ROUT method (*Motulsky and Brown, 2006*) (Q=1) and excluded from the analysis. The ROUT method is based on the false discovery rate (FDR), and the specified Q is the maximum desired FDR. See (*Motulsky and Brown, 2006*). The normality of sample distributions was assessed with the Shapiro-Wilk criterion, and when violated nonparametric tests were used. Nonparametric tests were also used when unequal variance was present. When normally distributed and with equal variance, the data were analyzed with unpaired t-tests, paired t-tests, and one-way ANOVA as appropriate. When normality was violated, the data were analyzed with Mann-Whitney test for unpaired test and Wilcoxon matched-pairs signed rank for paired test. For the analysis of variance with two factors (two-way ANOVA, two-way RM-ANOVA, or three-way RM-ANOVA), we first checked normality and the analysis was performed assuming no sphericity, which implies Geisser-Greenhouse correction. When the interaction between factors was present, the analysis was followed by Tukey's or Bonferroni's post hoc test as specified in each figure. For excitability experiments, multiple comparisons were made using uncorrected Fisher's LSD. All the statistical outputs are present in the 'source-data files.' Data are represented as the mean ± SEM, and the significance was set at 95% confidence interval. All the experimenters were blinded to perform behavioral manual scores and analyses.

## Acknowledgements

CB is supported by the Swiss National Science Foundation, Pierre Mercier Foundation, ERC consolidator grant, and NCCR Synapsy. The authors thank Lorena Jourdain for her technical support.

## Additional information

### Funding

| Funder | Grant reference number | Author |
| --- | --- | --- |
| Swiss National Science Foundation | 31003A_182326 | Camilla Bellone |

| Funder | Grant reference number | Author |
| --- | --- | --- |
| European Research Council | Consolidator grant 864552 | Camilla Bellone |
| NCCR Synapsy | 51NF40-185897 | Camilla Bellone |

The funders had no role in study design, data collection and interpretation, or the decision to submit the work for publication.

## Author contributions

Stefano Musardo, Conceptualization, Data curation, Formal analysis, Investigation, Methodology, Project administration, Software, Validation, Visualization, Writing - original draft, Writing – review and editing; Alessandro Contestabile, Formal analysis, Investigation, Visualization, Writing – review and editing; Marit Knoop, Olivier Baud, Formal analysis, Methodology, Writing – review and editing; Camilla Bellone, Conceptualization, Funding acquisition, Methodology, Writing - original draft, Writing – review and editing

## Author ORCIDs

Stefano Musardo ⓘD http://orcid.org/0000-0003-0392-0905
Alessandro Contestabile ⓘD http://orcid.org/0000-0001-6528-1403
Camilla Bellone ⓘD http://orcid.org/0000-0002-6774-6275

## Ethics

All the procedures performed at UNIGE complied with the Swiss National Institutional Guidelines on Animal Experimentation and were approved by the Swiss Cantonal Veterinary Office Committees for Animal Experimentation.

## Decision letter and Author response

Decision letter https://doi.org/10.7554/eLife.73421.sa1
Author response https://doi.org/10.7554/eLife.73421.sa2

# Additional files

## Supplementary files

• Transparent reporting form

## Data availability

All data generated or analysed during this study and the statistical results are included in the manuscript and supporting files. Source data files have been provided for all the figures and figure supplements.

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
