## [Editor Report]

This paper evaluated the lasting effects of acute social isolation on future social interactions in juvenile mice, revealing a compelling oxytocin-mediated mechanism. A clear hypothesis has been laid out within a defined anatomical framework, and social interactions were evaluated using appropriate behavioral paradigms, chemogenetic, and pharmacological tools. The work provides new insights on oxytocin signaling as a key regulator of the neural substrates underlying enduring effects of social interaction.

---

## [Decision Letter]

**Decision letter after peer review:**

Thank you for submitting your article "Oxytocin neurons control the effects of social isolation via the mesocortical pathway" for consideration by *eLife*. Your article has been reviewed by 3 peer reviewers, and the evaluation has been overseen by a Reviewing Editor and Kate Wassum as the Senior Editor. The following individual involved in review of your submission has agreed to reveal their identity: Michy Kelly (Reviewer #1).

The reviewers have discussed their reviews with one another, and the Reviewing Editor has drafted this to help you prepare a revised submission. In addition to addressing the concerns raised in the public review, please address each of the essential revisions provided below.

Essential revisions:

1) All reviewers noted that the lack of females profoundly lessen the impact of the findings. Ideally, all experiments would need to be conducted in females. However, the reviewers understand that this may not be feasible depending on the availability of animals and staffing levels. If this is the case, it is strongly recommended that the authors at the least acknowledge this significant shortcoming and add extensive discussion related to the effect of oxytocin in females (please see Caldwell, Curr. Op. Behavior. Sci, 2018).

2) There was consensus among the reviewers that the statistical analyses are not appropriate for the experimental design and this also needs to be addressed thoroughly. In particular:

a. Parametric statistics are based on assumptions of normality and equal variance, yet only a test of normality is mentioned and only for t-tests and one way ANOVAs. It is stated that normality was assumed for datasets requiring multifactorial ANOVAs. Please assess each dataset for both normality and equal variance before using parametric statistics. 2way ANOVAs frequently fail one or the other assumption, thus necessitating the employ of multiple non-parametric tests to assess the various factors (or, alternatively, a non-parametric multifactorial ANOVA if you have access to custom analysis software). This is a particular concern because in many datasets, variance appears to systematically increase with increasing mean (usually a sign of failed equal variance).

b. Throughout the manuscript, 2-way ANOVAs are reported for behavioral tasks where 2-way repeated measure ANOVAs should have been used (e.g., in the case of tasks tracking performance in the same subject across multiple compartments or objects). In the ephys data, 2 WAY RM ANOVA were used as appropriate; however, it is surprising there was not a significant interaction for many of those experiments (i.e., only main effects were reported) given the distinct overlap at "0". Was the "0" data not included in the RM ANOVA?

c. Throughout the manuscript, only main effects are reported when a significant interaction between effects would be required for the post hoc testing conducted and conclusions drawn. For example, in Figure 1G, the main effect of chamber that is reported says that across the experimental groups, novel > familiar. To statistically make a conclusion that the group vs isolated mice performed differently, you would need a significant interaction between chamber x housing condition to justify the post hoc tests examining the effect of chamber within group housed and chamber within isolated mice. In Figure 1J as well, you are trying to conclude the housing conditions perform differently across days. But to say that, the 2way rm would need to yield a significant interaction of housing condition x day to warrant a post hoc analysis of days within group housed and days within isolated. Again, this is not limited to Figure 1, I just use them here as examples.

d. Many 2-way ANOVAs have fractional df's reported. In the case of linear models, it is my understanding df's should be whole numbers, so it is not clear why dfs are presented in many 2way analyses as decimals (e.g., "2.027, 20.27). I believe in more complicated analyses, unequal variances can yield fractional df's, but in the case of unequal variance, a non-parametric statistic should be employed. Would you please clarify or revise?

e. The method of data presentation in Figure 1 S2 is not clear. Typically, "preference index" is a single number that incorporates relative preference for a novel object as a function of total time of exploration (=n-f/n+f). I do not believe it would be statistically appropriate to compare normalized data for ob1 vs ob2 because the combined value of both factors would be the same between groups (i.e., there could be no effect of group). The more typical way of analyzing these data is to compare the preference index for the novel object between the group vs. isolated mice to determine if there is a group effect, and then to conduct a one-sample t-test for each group's index vs.5 to determine if they exhibited significant memory (i.e., an index that differed significantly from chance). The authors may refer to the following paper for guidance:

https://www.sciencedirect.com/science/article/pii/S1074742714000070. If you were to analyze the raw seconds of exploration, then you could do a paired analyses of O1 vs O2, that said, the analysis should be a 2factor RM with housing condition vs object as factors. The same applies to the analyses of closed vs open arms.

f. Legend for Figure 3L is missing the post hoc analyses to say which group is different from which. Further, it seems equal variance could possibly fail given that variability increases in parallel with the increases mean (also with study in 3D).

g. In Figure 3 S1B, was there not a significant interaction? It seems there is an effect of increasing intensity within the no CNO group, but not the CNO group. Note that a significant interaction overrides main effects as it states that the effect of one factor depends on the level of the other factor.

3) It is not clear how oxytocin neurons were identified in Figure 3 S1. With current details provided, it appears that recordings of all PVN neurons (non-oxytocin neurons as well) were obtained in mice; please clarify. Moreover, cFOS+ and Oxt+ coexpression images are needed to relate the increase in the activity of PVN (via the IEG proxy) and the density of Oxt+ cells in PVN (figure 3 and page 4, line 11-17). Also, please clarify whether neurons are increasing Oxt production to become more efficiently labeled or changing identity of their synthesized neuropeptides?

4) Although it is reasonable to suggest that the lack of preference for the novel mouse and the failure to habituate to a mouse could reflect a reduced preference for novelty (and is that because they are less motivated to explore novel social stimuli or because they find social stimuli less rewarding?). This could also/alternatively reflect a social learning and/or memory deficit (they fail to form a memory of the first-presented mouse). To disentangle these seemingly opposite consequences, typically habituation sessions are conducted for a longer period of time and behavior examined across time within the same day (e.g., Hedge et al., 2016, Neuroscience https://www.ncbi.nlm.nih.gov/pmc/articles/PMC5031549/).

5) The extensive and definitive use of the term "social craving" throughout is unjustified based on the data presented (e.g., there is no breakpoint data shown). Craving implies having experienced something that is rewarding and wanting more of it; however, these mice never experienced any social interactions with non-cagemates, so how can they crave it? To this end, it might be informative to test these behaviors using cage mates as stimulus mice.

6) Regarding the statement "Indeed, we could speculate that social isolation during adolescence results in an intense urge to interact with whatever conspecific is present." This can be tested in a straightforward fashion by using cage mates as stimulus mice, or measuring interactions in the home cage, instead of using novel mice.

7) It is important to clarify why stimulus mice were 1 week younger and whether this is this true for adolescent and adult studies? This is of particular relevance because in the adult study illustrations, a small (seemingly juvenile) mouse is drawn--which would suggest much more than 1 week younger in the case of testing adults. If adults were tested with juvenile mice, as the illustration implies, it complicates interpretation of the adult isolation study since the differential effect of isolation may not be related to the period of isolation but rather a fundamentally different preference between adolescent and adult mice for other juvenile mice.

8) The authors identified putative DA neurons based on position, morphology, and capacitance, while providing sparse details in the methods. This section should contain detailed information and/or a small post hoc immunohistochemistry validated dataset.

9) The authors have exclusively shown scaled-up excitatory responses by pharmacologically blocking GABAA receptors, determining synaptic scaling after social isolation without considering potential effects on inhibitory post synaptic responses. Please provide prediction for the latter. Furthermore, it is not clear if synaptic scaling reported in Figure 5 induced by adolescent isolation only emerges slowly in the adult, or it is already acutely happening at the end of isolation at p35. Please discuss in detail.

10) It is unclear how oxytocin neuron activity contributes to synaptic plasticity onto VTA-NAc projections. One possibility is that oxytocin expressing PVN neurons projecting to VTA also release glutamate and contribute to the synaptic plasticity. It would be informative to optogenetically examine the synaptic connectivity between PVN->VTA projections and VTA->NAc projection neurons.

[Editors’ note: further revisions were suggested prior to acceptance, as described below.]

Thank you for resubmitting your work entitled "Oxytocin neurons mediate the effect of social isolation via the VTA circuits" for further consideration by *eLife*. Your revised article has been evaluated by Kate Wassum (Senior Editor) and a Reviewing Editor.

The manuscript has been improved but there are some remaining issues that need to be addressed, as outlined below:

1) Please acknowledge the lack of females as a shortcoming of the study and provide more discussion of this limitation; e.g., that differences in the system in males vs females and how these findings here may only apply to the male brain. Please also state the sex of the subjects in the abstract.

2) The post hoc analyses referred to in 2f of the letter are still missing. The statistical interaction was reported, but the post hoc tests were not, nor were significance markers added to the figure to indicate at which pA the groups differ.

3) Please make sure all statistics are reported. For instance, in some cases a 2 way ANOVA was used but only one main effect is reported.

https://reviewer.elifesciences.org/author-guide/full "Report exact p-values wherever possible alongside the summary statistics and 95% confidence intervals. These should be reported for all key questions and not only when the p-value is less than 0.05."

4) As isolated mice show increased social interaction during direct social interaction, but not during the 3 chamber sociability assay, it is important to more thoroughly explore why isolated mice show different outcomes between these two tests. To better understand the cause of increased direct social interaction (social craving vs aggression), it would be informative to conduct additional behavioral testings from isolated mice (e.g. social CPP, aggression assay). If new behavior testing data such as social CPP or aggression assay (resident intruder etc) is not possible, at the very least a more detailed behavioral characterization such as chasing during free social interaction test is needed to address this concern and better distinguish craving vs aggressive encounter. Overall, more careful consideration is required to interpret the nature social behavior in isolated mice captured by free social interaction and 3 chamber task.

5) The response to essential review 2c to statistically compare between groups was not fully addressed. [Results] page 1, lines 18-19 and Figure 1EF. To evaluate if the house condition modifies the sociability, it is essential to run these statistics for Figure 1HI. Related to this, the 3 chamber testing data should be presented and analyzed in a similar way across different sets of data. E.g., Figure 1 HI are missing center chamber time and the preference index data while all the rest of the 3 chamber data include these.

In general, more clarity on whether or not the 3-chamber social preference test results do or do not support the conclusion of social isolation leading to an increase in social interaction would help. E.g., by clarifying whether isolation leads to an increase in social interaction specifically in "free/unrestrained"social interactions in adulthood.

---

## [Author Response]

Essential revisions:1) All reviewers noted that the lack of females profoundly lessen the impact of the findings. Ideally, all experiments would need to be conducted in females. However, the reviewers understand that this may not be feasible depending on the availability of animals and staffing levels. If this is the case, it is strongly recommended that the authors at the least acknowledge this significant shortcoming and add extensive discussion related to the effect of oxytocin in females (please see Caldwell, Curr. Op. Behavior. Sci, 2018).

We thank the reviewers for this comment. We agree that results from the female cohort would increase the impact of the findings; however, these experiments would take a considerable amount of time and be a matter of future study. As suggested by the reviewers, we have now added a comment about this point (p 9 lines 8 to 15)

2) There was consensus among the reviewers that the statistical analyses are not appropriate for the experimental design and this also needs to be addressed thoroughly. In particular:

We apologize for the inappropriate statistical analysis. In the new version of the paper we have revised all the statistical analysis and for each “Source data” file, we have now added the output of all statistical tests that were used.

a. Parametric statistics are based on assumptions of normality and equal variance, yet only a test of normality is mentioned and only for t-tests and one way ANOVAs. It is stated that normality was assumed for datasets requiring multifactorial ANOVAs. Please assess each dataset for both normality and equal variance before using parametric statistics. 2way ANOVAs frequently fail one or the other assumption, thus necessitating the employ of multiple non-parametric tests to assess the various factors (or, alternatively, a non-parametric multifactorial ANOVA if you have access to custom analysis software). This is a particular concern because in many datasets, variance appears to systematically increase with increasing mean (usually a sign of failed equal variance).

We thank the reviewer for the comment. We now used a parametric test only if both normality and equal variance were not violated in the revised statistical analysis. If the contrary, we used a non-parametric test. Regarding multifactorial ANOVAs, we first checked normality, and all the analysis were then performed assuming no sphericity, which implies Geisser-Greenhouse correction. We have now included these details in Material and methods, and all the stats have been upgraded in the figures and legends.

b. Throughout the manuscript, 2-way ANOVAs are reported for behavioral tasks where 2-way repeated measure ANOVAs should have been used (e.g., in the case of tasks tracking performance in the same subject across multiple compartments or objects). In the ephys data, 2 WAY RM ANOVA were used as appropriate; however, it is surprising there was not a significant interaction for many of those experiments (i.e., only main effects were reported) given the distinct overlap at "0". Was the "0" data not included in the RM ANOVA?

We apologize for the mistake, and we have now used the appropriate statistical test for each dataset. In all the figures we have now reported the F value for the “interaction” between variables, and as written before, each “Data source” file contains all statistical analysis outputs.

c. Throughout the manuscript, only main effects are reported when a significant interaction between effects would be required for the post hoc testing conducted and conclusions drawn. For example, in Figure 1G, the main effect of chamber that is reported says that across the experimental groups, novel > familiar. To statistically make a conclusion that the group vs isolated mice performed differently, you would need a significant interaction between chamber x housing condition to justify the post hoc tests examining the effect of chamber within group housed and chamber within isolated mice. In Figure 1J as well, you are trying to conclude the housing conditions perform differently across days. But to say that, the 2way rm would need to yield a significant interaction of housing condition x day to warrant a post hoc analysis of days within group housed and days within isolated. Again, this is not limited to Figure 1, I just use them here as examples.

We have now modified the statistical analysis according to what we have recently published (Mol Psychiatry. 2022 Jan 12. doi: 10.1038/s41380-021-01427-0; Eur J Neurosci. 2021 May;53(9):3199-3211. doi: 10.1111/ejn.15179). Since the aim of the 3-chamber task is to evaluate if a mouse prefers a social stimulus over an object and a novel social stimulus over a familiar one, we decided to perform a stand-alone analysis for each group of mice. In particular, we used paired t-test analysis for interaction time (new Figure1 H-I and new Figure1-Supplement 1E, G), One-way RM ANOVA for time in the chamber (new Figure1-Supplement 1D, F), and unpaired samples t-test for the distance moved (new Figure1-Supplement 1C). The 3-chamber task for Figure1-Supplement 2 and Figure4-Supplement 1 were analyzed in the same way.

Regarding the habituation task (new Figure1J-N, new Figure 4E-I, and new Figure1-Supplement 1H-L ), we now have included the interaction effect for the 2-way RM ANOVA and we added new panels comparing the interaction time between DAY1 (or TRIAL1) and DAY4 (or TRIAL4).

d. Many 2-way ANOVAs have fractional df's reported. In the case of linear models, it is my understanding df's should be whole numbers, so it is not clear why dfs are presented in many 2way analyses as decimals (e.g., "2.027, 20.27). I believe in more complicated analyses, unequal variances can yield fractional df's, but in the case of unequal variance, a non-parametric statistic should be employed. Would you please clarify or revise?

We thank the reviewer for this comment. The presence of fractional df’s is due to the Geisser-Greenhouse correction. We now added all the statistical details in the material and methods section.

e. The method of data presentation in Figure 1 S2 is not clear. Typically, "preference index" is a single number that incorporates relative preference for a novel object as a function of total time of exploration (=n-f/n+f). I do not believe it would be statistically appropriate to compare normalized data for ob1 vs ob2 because the combined value of both factors would be the same between groups (i.e., there could be no effect of group). The more typical way of analyzing these data is to compare the preference index for the novel object between the group vs. isolated mice to determine if there is a group effect, and then to conduct a one-sample t-test for each group's index vs.5 to determine if they exhibited significant memory (i.e., an index that differed significantly from chance). The authors may refer to the following paper for guidance:https://www.sciencedirect.com/science/article/pii/S1074742714000070. If you were to analyze the raw seconds of exploration, then you could do a paired analyses of O1 vs O2, that said, the analysis should be a 2factor RM with housing condition vs object as factors. The same applies to the analyses of closed vs open arms.

We thank the reviewer for the comment; however we would like to point out that using “n/(n+f)” (the one that we use in our paper) or “(n-f)/(n+f)” as preference index will give the same statistically results. Indeed, the latter is a mathematical transposition of the first equation. However, since a mouse has a 50% of chance to explore one or the other object, we believe that our way to calculate preference index is more intuitive because if a mouse explores for the same amount of time the two objects, the preference index will be equal to 0.5, which correspond to the chance level. Our analysis aimed to demonstrate that both groups of mice were able to prefer a novel object over a familiar one and not to state that one group performed better compared to the other. Indeed, we first show no bias toward one of the two objects during the familiarization phase. Certainly, the paired t-test between the preference index for objects one and two is not significantly different. Our analysis mathematically gives the same result as calculating a one-sample t-test against the chance level. Since the social novelty was impaired in the three-chamber task in social isolated mice, our analysis of the NORT wants to determine if isolated mice discriminate a novel versus a familiar object and not compare performances between groups. We applied the same analysis for the EPM.

f. Legend for Figure 3L is missing the post hoc analyses to say which group is different from which. Further, it seems equal variance could possibly fail given that variability increases in parallel with the increases mean (also with study in 3D).

We revised the statistical analysis, and we decided to use a three-way RM ANOVA since we have three variables (current step, house condition, and CNO treatment). Moreover, we slightly increase the n for some groups to have a more homogeneous number.

g. In Figure 3 S1B, was there not a significant interaction? It seems there is an effect of increasing intensity within the no CNO group, but not the CNO group. Note that a significant interaction overrides main effects as it states that the effect of one factor depends on the level of the other factor.

We are sorry for the lack of information. We now added all the statistical analysis details.

3) It is not clear how oxytocin neurons were identified in Figure 3 S1. With current details provided, it appears that recordings of all PVN neurons (non-oxytocin neurons as well) were obtained in mice; please clarify. Moreover, cFOS+ and Oxt+ coexpression images are needed to relate the increase in the activity of PVN (via the IEG proxy) and the density of Oxt+ cells in PVN (figure 3 and page 4, line 11-17). Also, please clarify whether neurons are increasing Oxt production to become more efficiently labeled or changing identity of their synthesized neuropeptides?

We are sorry for the lack of information in the previous version of the manuscript. We have now added in the material and methods section the relative details. The OXT cells expressing hM4Di channel can be recognized thanks to the mCitrine reporter present on the hM4Di transgene Thus, in new Figure 3S1 J-L, we report excitability recorded from identified OXT neurons. Moreover, in the new figure 3-S1, we now added the immunohistochemical validation of the OXT-hM4Di mouse line (new figure 3-S1I).

For the cFOS/OXT analysis, we have now repeated the experiment, and we performed triple staining with three different antibodies against cFOS, OXT, and AVP, and analyzed the relative and absolute expression of the three proteins as well as the co-expression of cFOS-OXT, cFOS-AVP and OXT-AVP (new Figure 3A-D and new figure 3-S1A-G). Our new analysis shows an increase in the absolute number of OXT+ and cFOS + cells in isolated mice and an increase in the double-positive cells (OXT+ /cFOS + cells). Furthermore, since no differences were found in AVP expression or OXT-AVP colocalization, the results suggest that PVN neurons from isolated mice produced oxytocin more efficiently.

4) Although it is reasonable to suggest that the lack of preference for the novel mouse and the failure to habituate to a mouse could reflect a reduced preference for novelty (and is that because they are less motivated to explore novel social stimuli or because they find social stimuli less rewarding?). This could also/alternatively reflect a social learning and/or memory deficit (they fail to form a memory of the first-presented mouse). To disentangle these seemingly opposite consequences, typically habituation sessions are conducted for a longer period of time and behavior examined across time within the same day (e.g., Hedge et al., 2016, Neuroscience https://www.ncbi.nlm.nih.gov/pmc/articles/PMC5031549/).

We thank the reviewer for the comment. To prove that social isolation does not induce social learning or memory deficits, we performed another habituation task presenting the same social stimulus for 4 consecutive trials for 2 minutes with an intertrial interval of 5 minutes. We scored the interaction time for each trial. We observed that both Grouped and Isolated mice habituate to the same social stimulus with these settings. Indeed, the interaction time on Trial 4 is statistically different from Trial 1 for both groups (new figure 1-S1I-M). These results demonstrate that isolated mice can form short-time memory of the social stimulus while long term memory is impaired (new figure 1I-M)

5) The extensive and definitive use of the term "social craving" throughout is unjustified based on the data presented (e.g., there is no breakpoint data shown). Craving implies having experienced something that is rewarding and wanting more of it; however, these mice never experienced any social interactions with non-cagemates, so how can they crave it? To this end, it might be informative to test these behaviors using cage mates as stimulus mice.6) Regarding the statement "Indeed, we could speculate that social isolation during adolescence results in an intense urge to interact with whatever conspecific is present." This can be tested in a straightforward fashion by using cage mates as stimulus mice, or measuring interactions in the home cage, instead of using novel mice.

We thank the reviewer for the comment. We now performed free-social interaction after social isolation using former cage mates as stimuli as suggested by the reviewer (new figure 1E-F). We found that isolated mice interact more compared to grouped mice even when stimulus mice are former cage-mate.

7) It is important to clarify why stimulus mice were 1 week younger and whether this is this true for adolescent and adult studies? This is of particular relevance because in the adult study illustrations, a small (seemingly juvenile) mouse is drawn--which would suggest much more than 1 week younger in the case of testing adults. If adults were tested with juvenile mice, as the illustration implies, it complicates interpretation of the adult isolation study since the differential effect of isolation may not be related to the period of isolation but rather a fundamentally different preference between adolescent and adult mice for other juvenile mice.

We thank the reviewer for the comment. As stated in the material methods, stimuli mice are always 1-week younger compared to the experimental mice thus they are P28 for adolescent experimental mice, and P50-53 for adult experimental mice. We have now modified the illustration in figure 1-S2B to be more in line with the age of the mice.

8) The authors identified putative DA neurons based on position, morphology, and capacitance, while providing sparse details in the methods. This section should contain detailed information and/or a small post hoc immunohistochemistry validated dataset.

We are sorry for the lack of information. We have provided all the details now in the material and method session. Moreover, we now added a post-hoc immunohistochemistry validation (new figure 5 – Supplement 2)

9) The authors have exclusively shown scaled-up excitatory responses by pharmacologically blocking GABAA receptors, determining synaptic scaling after social isolation without considering potential effects on inhibitory post synaptic responses. Please provide prediction for the latter. Furthermore, it is not clear if synaptic scaling reported in Figure 5 induced by adolescent isolation only emerges slowly in the adult, or it is already acutely happening at the end of isolation at p35. Please discuss in detail.

We thank the reviewer for the comment. We have now performed new experiments and patched DA neurons projecting the PFC or NAc obtained from adolescent mice immediately after the social isolation. We found an increase in rectification index when recoding from DA neurons projecting to PFC at P35 (new figure 5-supplement 1A-K). Moreover, we now recorded sIPSCs from VTA DA neurons, and we found no differences in amplitude, frequency, rise, and decay time of the inhibitory postsynaptic currents, suggesting thus that inhibitory responses are not affected by adolescent social isolation.

10) It is unclear how oxytocin neuron activity contributes to synaptic plasticity onto VTA-NAc projections. One possibility is that oxytocin expressing PVN neurons projecting to VTA also release glutamate and contribute to the synaptic plasticity. It would be informative to optogenetically examine the synaptic connectivity between PVN->VTA projections and VTA->NAc projection neurons.

We thank the reviewer for the comment. As suggested by the reviewer, we injected ChR2 in the PVN at PND5 and CholeraToxin in the mPFC at PND21. We then isolated the mice for seven days (P28 and P35), and we recorded from DA neurons projecting to the PFC while optogenetically stimulating PVN inputs in slices. Unfortunately, the injection of ChR2 in the PVN at PND5 resulted very challenging and we were not able to measure any current due to the lack of infection of the PVN.

[Editors’ note: further revisions were suggested prior to acceptance, as described below.]

1) Please acknowledge the lack of females as a shortcoming of the study and provide more discussion of this limitation; e.g., that differences in the system in males vs females and how these findings here may only apply to the male brain. Please also state the sex of the subjects in the abstract.

We thank the reviewer for the comment. We have now clearly stated in the abstract that we used only male, and we add a discussion about the limitation of the study in the discussion session (lines265-275).

2) The post hoc analyses referred to in 2f of the letter are still missing. The statistical interaction was reported, but the post hoc tests were not, nor were significance markers added to the figure to indicate at which pA the groups differ.

We are sorry for the lack of information. All the statistical parameters are now reported in the legend. Post-hoc analyses have now been reported. For figure 3K (former figure 3L) we reported in the figure the post-hoc analysis of isolated group versus all other groups while reported in the source file all the rest of the comparison.

3) Please make sure all statistics are reported. For instance, in some cases a 2 way ANOVA was used but only one main effect is reported. https://reviewer.elifesciences.org/author-guide/full "Report exact p-values wherever possible alongside the summary statistics and 95% confidence intervals. These should be reported for all key questions and not only when the p-value is less than 0.05."

We are sorry for the lack of statistical details that we have now added all the statistical information in the revised version of the manuscript. Moreover, all the statistical data are also reported in the source files.

4) As isolated mice show increased social interaction during direct social interaction, but not during the 3 chamber sociability assay, it is important to more thoroughly explore why isolated mice show different outcomes between these two tests. To better understand the cause of increased direct social interaction (social craving vs aggression), it would be informative to conduct additional behavioral testings from isolated mice (e.g. social CPP, aggression assay). If new behavior testing data such as social CPP or aggression assay (resident intruder etc) is not possible, at the very least a more detailed behavioral characterization such as chasing during free social interaction test is needed to address this concern and better distinguish craving vs aggressive encounter. Overall, more careful consideration is required to interpret the nature social behavior in isolated mice captured by free social interaction and 3 chamber task.

We thank the reviewer for the comment. As suggested, we performed the social conditioning place preference task after isolation. The new results (Figure 1 – Supplement 2 M-R) show that both Grouped and Isolated mice show preference for the compartment paired with social interaction. Concerning figure 1H, since the ANOVA does not show interaction for social preference, we are not allowed to perform multiple comparison. However, we would like to report that if we perform an unpaired t-test between interaction time stimulus 1 for grouped and isolated mice, we found a significant difference (t_(30)_=2.167, p=0.0383).

5) The response to essential review 2c to statistically compare between groups was not fully addressed. [Results] page 1, lines 18-19 and Figure 1EF. To evaluate if the house condition modifies the sociability, it is essential to run these statistics for Figure 1HI. Related to this, the 3 chamber testing data should be presented and analyzed in a similar way across different sets of data. E.g., Figure 1 HI are missing center chamber time and the preference index data while all the rest of the 3 chamber data include these.In general, more clarity on whether or not the 3-chamber social preference test results do or do not support the conclusion of social isolation leading to an increase in social interaction would help. E.g., by clarifying whether isolation leads to an increase in social interaction specifically in "free/unrestrained"social interactions in adulthood.

We thank the reviewer for the comment. The data regarding the time in chamber and preference index for figure 1 were already present in Figure 1 – supplementary 1. Since figure 1 has many panels, we decided to include in the main figure only the interaction time.

In regards of the statistical analysis for the three-chamber test throughout the manuscript, in this new version of the manuscript we revised all the statistic such as:

– For the interaction time, we first performed a paired t-test within single groups. This statistical test is important for the sociability test to verify that control group prefer stimulus over object. Indeed, the absence of sociability in this group would drop all conclusions. As now requested by the reviewer, we also performed RM 2-way ANOVA. Now all the statistical analysis have been reported in the figures, legends, and source data files. Multiple comparisons were made when interaction between variables was significant.

– For time in chamber, One-way RM ANOVA

– For preference index, One-sample t-test against chance level.

We now show that during the social novelty phase (figure 1I) there is an interaction between the house condition and the interaction time confirming that social isolation induces an alteration of the social novelty preference.

Moreover, as suggested, we added in the text “in free/unrestrained social interaction” to better clarify the effect of social isolation in adulthood.